# Identification and structural analysis of the tripartite α-pore forming toxin of *Aeromonas hydrophila*

Jason S. Wilson [1,4], Alicia M. Churchill-Angus [1,4], Simon P. Davies [1,2], Svetlana E. Sedelnikova[1], Svetomir B. Tzokov [1], John B. Rafferty [1], Per A. Bullough[1], Claudine Bisson [1,3] & Patrick J. Baker [1]

The alpha helical CytolysinA family of pore forming toxins (α-PFT) contains single, two, and three component members. Structures of the single component *Eschericia coli* ClyA and the two component *Yersinia enterolytica* YaxAB show both undergo conformational changes from soluble to pore forms, and oligomerization to produce the active pore. Here we identify tripartite α-PFTs in pathogenic Gram negative bacteria, including *Aeromonas hydrophila* (AhlABC). We show that the AhlABC toxin requires all three components for maximal cell lysis. We present structures of pore components which describe a bi-fold hinge mechanism for soluble to pore transition in AhlB and a contrasting tetrameric assembly employed by soluble AhlC to hide their hydrophobic membrane associated residues. We propose a model of pore assembly where the AhlC tetramer dissociates, binds a single membrane leaflet, recruits AhlB promoting soluble to pore transition, prior to AhlA binding to form the active hydrophilic lined pore.

[1] Department of Molecular Biology and Biotechnology, University of Sheffield, Firth Court, Western Bank, Sheffield, South Yorkshire S10 2TN, UK. [2] School of Biomedical Sciences, Faculty of Biological Sciences and Astbury Centre for Structural and Molecular Biology, University of Leeds, Leeds LS2 9JT, UK. [3] ISMB, Department of Biological Sciences, Birkbeck, University of London, Malet Street, London WC1E 7HX, UK. [4]These authors contributed equally: Jason Wilson, Alicia Churchill-Angus. Correspondence and requests for materials should be addressed to P.J.B. (email: p.baker@sheffield.ac.uk)

Pore forming toxins (PFTs) are critical components of the molecular offensive and defensive machinery of cells in virtually all kingdoms of life. In eukaryotes, PFTs are largely involved in the innate immune response[1], whilst in bacteria, PFTs form the major group of virulence factors in many pathogenic bacteria and constitute 30% of all toxins identified to date[2]. By puncturing holes in the membrane, bacterial PFTs facilitate the takeover of host resources, destroy the osmotic balance of the target cell[3] or insert a secondary intracellular toxin (such as with the anthrax binary toxin), through the pore formed in the membrane[4]. Many pathogenic bacteria, including highly anti-biotic resistant strains, employ PFTs in their invasive arsenal. This makes them an attractive target for the development of virulence-targeted therapies that may have broad spectrum activity against human and other pathogens, a strategy that has potential to reduce the acquired resistance seen in conventional antimicrobial therapy[5].

All PFTs are produced as a soluble, generally monomeric form, which recognises the target cell by binding to specific receptors, thus concentrating the proteins to the membrane surface, before exposure of the transmembrane hydrophobic regions, oligomer-ization, and membrane insertion[6,7]. Depending on the secondary structure of the membrane component, PFTs can be classified as α-PFTs, using a ring of amphipathic helices to construct the pore or as β-PFTs, where a β-barrel is used to traverse the membrane[7].

Although a number of structures of α-PFTs have been deter-mined by X-ray crystallography and electron microscopy[7], few are known in both pore and soluble form. One α-PFT where this transition is well understood is Cytolysin A (ClyA) from enter-opathogenic *Escherichia coli* and *Salmonella* species[8,9]. In this protein a large conformational change occurs, involving 80% of residues, as the single soluble subunit rearranges to expose its membrane spanning hydrophobic residues (the β-tongue) and oligomerises to form the active pore of 6–13 subunits[9,10]. In this pore the hydrophobic β-tongue refolds to form the ends of two adjacent helices and these insert into a single leaflet of the target bilayer, with the amphipathic N-terminal helix providing the membrane spanning component and thus a hydrophilic lined pore. A number of α-PFTs have been identified which show structural similarity to ClyA but are composed of two protein components (bipartite PFTs). These include YaxAB from the human pathogenic bacteria *Yersinia enterolitica*[11,12], the PaxAB system of the insect pathogen *Photorhabdus luminescens*[11,13] and the XaxAB system from the insect pathogen *Xenorhabdus nematophila*[14,15]. The structures of the YaxAB and XaxAB pores and their component parts have been determined[11,15], and all show structural similarity to ClyA, but with low-sequence identity (approx 22% identity)[11]. In the active YaxAB pore the two pro-teins are arranged as ten heterodimers in a pore with C10 symmetry, while the XaxAB pore shows variability in pore protomer count, with 12–15 heterodimers observed. These bipartate pores have a pronounced funnel shape compared to the largely cylindrical ClyA pore due to a more extensive extracellular component[11,15], with the A component attaching to a single leaflet and the B component forming the membrane spanning region using two amphipathic helices. During pore formation the membrane spanning residues in YaxB and XaxB are exposed by a conformational change that is smaller in extent, but similar in effect, to that seen in ClyA[10,11,15].

Two α-PFT homologues to ClyA have also been identified in pathogenic strains of *Bacillus cereus*, but in these systems three proteins encoded on the same operon are required for maximal haemolytic activity (tripartite PFTs). One of these α-PFTs in *B. cereus* is formed from the Hbl-L$_1$, Hbl-L$_2$ and Hbl-B proteins[16–19], with the second comprised of proteins NheA, NheB and NheC[20,21]. In the latter system, NheC primes the host cell for the formation of ion permeable NheB/C pores[22], prior to the addition of NheA to construct the complete pore. Structures of the soluble NheA and HblB monomers are known, which both show similarity to ClyA, but again with less than 20% sequence identity[23,24]. As yet, the mode of assembly of this tripartite class of ClyA family α-PFTs is unknown.

Here, we identify tripartite ClyA family toxins in the genomes of medically and economically important pathogenic Gram-negative bacteria, including *Serratia marcescens*, a causative agent of nosocomial infections[25]; *Erwinia mallotivora*, responsible for papaya dieback disease[26], resulting in $58 million/annum damage to crops in Malaysia; and highly virulent strains of *Aeromonas hydrophila*, which cause fatal haemorrhagic septicaemia in a number of important farmed fish species with epidemics in US and Asian aquaculture causing multimillion dollar losses[27,28], as well as being a contributor to opportunistic human disease[29]. We show that the orthologue identified in *A. hydrophila* (AhlABC) is a haemolytic tripartite α-PFT, requiring all three components for maximal lysis, and we present the structures of the soluble forms of monomeric AhlB and tetrameric AhlC and the decameric pore form of AhlB. Our data presented here has allowed the con-struction of a model to show how a three component α-PFT system could assemble to create a lethal pore.

We show, guided by functional and mutagenic studies, that the AhlC tetramer first disassembles into monomers in order to form the initial membrane-binding event. We also show that AhlB undergoes a large conformational change similar to ClyA YaxB and XaxB[10,11,15], involving the beta tongue becoming an exten-ded alpha helix and the tail domain helices sliding relative to one another as AhlB assembles into a hydrophobic pore. Secondary structure modelling of the predicted head domain of AhlA shows that similar conformational changes would expose an amphi-pathic helix, which on assembly of the AhlABC pore could pro-vide a hydrophilic lining to the pore as seen in the other family members ClyA, YaxAB and XaxAB[10,11,15] .

## Results

**A. hydrophila has a tripartite α-PFT**. Although single and two component α-PFTs have been identified, the only known three component ClyA family members are the tripartite NheABC and HblL$_1$L$_2$B systems of *B. cereus*. We thus instigated a bioinfor-matics search based on both the sequence and structure of the NheA, NheB and NheC proteins to discover further tripartite α-PFTs. The hypothetical protein translated from the AXH33180.1 gene of *A. hydrophila* species AL09-71, was identified with 24% sequence identity and 46% similarity, as defined by TCoffee[30], to NheB. The proteins coded by the upstream and downstream adjacent genes to AXH33180.1 in the *A. hydrophila* genome (AHX33179.1 and AHX33181.1, respectively) had only 6% and 9% sequence identity to NheA and NheC, yet their sequence similarity was 39% and 37%, respectively. Furthermore, hydro-pathy plots of these three *A. hydrophila* gene products showed a very similar pattern of hydrophobic regions as those seen in the NheABC tripartite toxin (Supplementary Fig. 1). The AHX33180.1 gene product contained a possible membrane-spanning region of 63 hydrophobic residues, similar in length and position to the 54 residue predicted membrane spanning region of NheB. Similarly, the AHX33181.1 gene product had a stretch of 23 hydrophobic residues in the same place in the sequence as seen for 22 hydrophobic residues in NheC. In addition, the sequence of the AHX33179.1 gene product was devoid of sig-nificant stretches of hydrophobic residues, a pattern also observed in NheA (Supplementary Fig. 1). These three *A. hydrophila* proteins were thus provisionally identified as a tripartite α-PFT and named AhlA, AhlB and AhlC. The AhlA and AhlB sequences

have since been annotated in NCBI as having a HBL type fold (PFAM05791).

To expand the family further, the three genes identified in *A. hydrophila* were used to identify orthologues in other Gram-negative bacteria. A total of 25 gene products with homology to the AhlA and 36 gene products with homology to AhlC were identified with *E* values < 0.01, including the known bipartite Gram-negative α-PFT components YaxA, PaxA and XaxA, whereas over fifty gene products were identified with homology to AhlB (Supplementary Data 1) suggesting that numerous α-PFT systems may exist in Gram-negative bacteria. Seven examples of full tripartite α-PFT systems were identified, including those in three human pathogens (*Serratia marcescens*, *Serratia liquefaciens* and *Spirosoma fluviale*), and two plant pathogens (*Erwinia mallotivora* and *Chromobacterium piscinae*) (Supplementary Table 1).

**All three AHL components are required for maximum AHL lysis.** Having identified that the AhlABC system of *A. hydrophila* could potentially be a tripartite PFT, each component was cloned, expressed and purified and used in haemolytic assays with horse erythrocytes (Fig. 1a, b). In isolation, each of the Ahl components exhibited no haemolytic activity. When tested in 1:1 binary combinations the AhlB/AhlC mixture showed 50% of the lysis observed when cells were lysed by osmotic shock with water, whereas the AhlA/AhlB and AhlA/AhlC mixtures exhibited no activity. Size exclusion chromatography of AhlB mixed with an excess of AhlC in aqueous solution did not show any AhlB/AhlC association (Supplementary Figure 2), indicating that any complex formation of these two components of *A. hydrophila* only occurs on the membrane. This is in contrast to both YaxAB and XaxAB where the A and B components are seen to form a complex before binding the membrane[11,15] and also for the *B. cereus* tripartite PFT where equimolar amounts of NheB and NheC produced an inhibitory complex in solution that prevented cell lysis[20,31]. A 1:1:1 mixture of all three Ahl components showed 95% of the lysis of the positive control, indicating that all three components of the AhlA/AhlB/AhlC toxin system are required for maximal lysis.

To elucidate the effect of the concentration of each protein on lytic activity, the ratio of individual proteins was varied within a mixture of AhlA, AhlB and AhlC by serial dilutions (Fig. 1b). When AhlA was diluted by one-half, lytic activity was reduced by 50%, to a level similar to that seen for AhlB and AhlC. However, when AhlC was diluted eightfold in the AhlA/AhlB/AhlC mixture 80% of the haemolytic activity remained. In contrast, an increasing reduction in lytic activity was seen across the dilution series when AhlB was diluted in the AhlA/AhlB/AhlC mixture. These results indicate that maximal lysis requires equal amounts of AhlA, AhlB and AhlC, and decreasing the concentration of any one of these components reduces lytic activity, with the most dramatic effects seen for AhlB and AhlA. Furthermore, when all three proteins are present at equal concentrations or increased above 1:1:1 no inhibitory effects are seen, in contrast to the *B. cereus* NheABC toxin where concentrations of NheC above a NheA:NheB:NheC ratio of 10:10:1 are inhibitory[20].

**Pre-incubation of erythrocytes with AhlC promotes rapid lysis.** To determine the binding order of each protein, time course assays were run using erythrocytes incubated with one or two of the Ahl components before addition of the other components (Fig. 1c, d). When AhlA/AhlB/AhlC, or AhlB/AhlC, were incubated together with erythrocytes there was a lag time of 20 min before lysis. However, if the cells were pre-incubated with AhlC before addition of AhlA and AhlB, the lag time was removed (or

reduced if just AhlB was added) indicating that AhlC may prime the membrane for pore assembly. In contrast, cells pre-incubated with AhlB prior to the addition of AhlC showed reduced lysis compared to cells incubated with AhlB and AhlC together, suggesting AhlB is inhibitory in the presence of a membrane, similar to that observed in the Nhe PFT, where pre-incubation of NheB with vero cells prevented formation of a functional complex[31]. Cells pre-incubated with AhlB and AhlC, prior to the addition of AhlA, showed a lag time, with lysis not reaching that of cells incubated with AhlA/AhlB/AhlC together, whereas removal of the lag time and maximal lysis occurred for erythrocytes pre-incubated with AhlA, prior to addition of AhlB and AhlC, showing that addition of AhlA to a pre-formed AhlB/AhlC pore is slower than if the full tripartite pore can assemble with all three components present. The end points of these time course assays were viewed by negative stain electron microscopy. Pores could only be seen in erythrocyte membranes incubated with AhlB, AhlB/AhlC or AhlA/AhlB/AhlC (Fig. 1e). The diameter of the AhlB pores was estimated from 100 particles to be 10 nm (s.d. = 2 nm), with the AhlB/AhlC and AhlA/AhlB/AhlC pores appearing significantly larger with a diameter of 16 nm (*n* = 100, s.d. = 2 nm), a similar size to that of the widest head region of the YaxAB and XaxAB pores[11,15].

The interaction between the Ahl components with lipid bilayers was investigated using both liposome float assays and ultracentrifugation in the presence of detergent to separate large assemblies, (>400 kDa) from soluble protein components (Fig. 1f, g). AhlA, AhlB and AhlC together bound to the liposomes with negative stain EM images showing that pores had been formed. Incubation of all three Ahl components with detergent showed they were also present in the large assemblies isolated by ultracentrifugation, with pores identified by negative stain EM (Supplementary Fig. 3). Similarly, pores and large assemblies could be identified for both AhlB and AhlB/AhlC in both liposomes and by ultracentrifugation. No pores were seen for AhlA/AhlC together, or separately, despite both these components binding to liposomes and no protein was present in the respective ultracentrifugation pellets, which implies that the pores observed for the AhlA/AhlB/AhlC mixture contained all three PFT components, with equal amounts present despite AhlB and AhlC being in excess. As no pores could be identified in negative stain EM images of the float assay control fractions without liposomes (Supplementary Fig. 3), we concluded that large complexes of both AhlB/AhlC and AhlA/AhlB/AhlC can only be formed in the presence of detergent or lipid bilayers, with a preference for the AhlABC complexes and that large oligomers of AhlB can be formed under the same conditions.

Negative stain EM showed that when AhlC was incubated together with AhlB in liposome or erythrocyte preparations, there was a dramatic increase in the number of pore-like structures in individual liposomes, compared to the situation with AhlB alone, where only a few pores were seen (Fig. 1e), indicating that AhlC modulates AhlB association with lipid. As sufficient protein was present for pores to be present in all liposomes it is interesting to note that some liposomes were saturated with AhlBC pores and others remained free of pores (Supplementary Fig. 3), suggesting cooperativity in pore formation. Indeed, when AhlA, AhlB and AhlC were incubated together all liposomes were saturated, a situation also observed for ClyA[32].

**Structure of the soluble form of AhlB.** The structure of the soluble form of AhlB was determined to 2.3 Å resolution (Table 1, Supplementary Fig. 4a). AhlB folds into a compact five helical bundle structure (α1–α5), with an associated domain constructed from a mix of three alpha helices and three beta strands. The

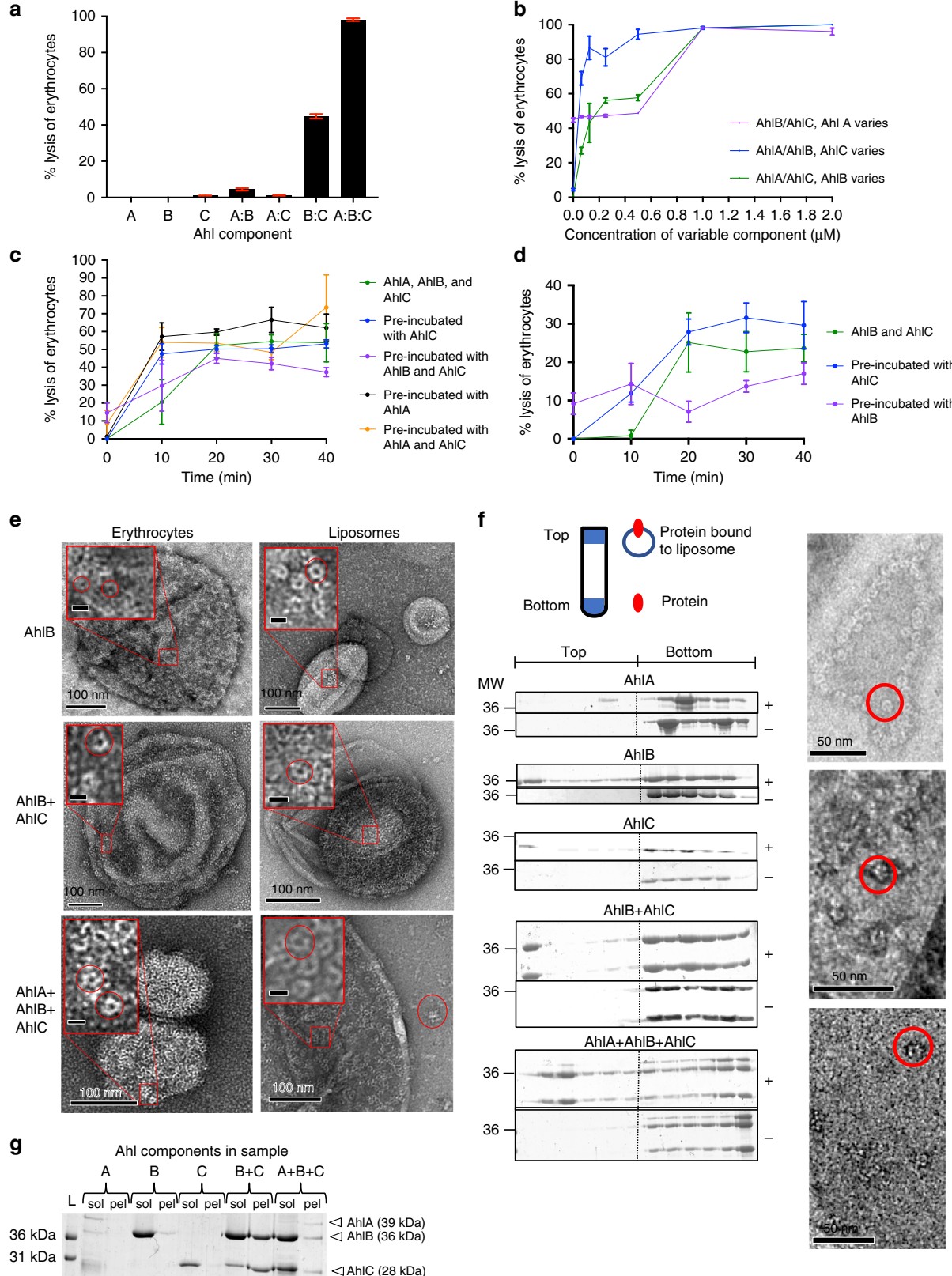

overall structure of soluble AhlB is very similar to that seen in the soluble forms of ClyA (pdb:1QOY), MakA (pdb:6EZV), NheA (pdb: 4K1P) and HblB (pdb:2NRJ) (Fig. 2a). Within this compact soluble structure, residues G191–A222, identified from hydropathy plots as a hydrophobic domain that possibly inserts into the membrane (Supplementary Fig. 1), fold into two antiparallel β-

strands (β1 and β2; the β-tongue) (Fig. 2b). These hydrophobic residues are thus shielded from solvent by packing against residues of helix α4 (E254–V270), α3a (residues V170–L186), α4a (residues G225–G239), the N-terminal helix (α1, residue G12–V34) and the C-terminus (residues T331–A359), which forms a short beta strand (β3) and helix (α5a). In this soluble

**Fig. 1** Lytic activity of the AHL toxin against horse erythrocytes. **a** Percentage lysis when each protein was incubated with horse erythrocytes alone and in combination at 1:1 ratios for 1 h. **b** % Lysis against varying AhlA, AhlB or AhlC concentration, when all three proteins were incubated together for 1 h. The other two proteins were at a fixed concentration of 1 μM. **c** % lysis against time when erythrocytes were incubated with 500 nM AhlA, AhlB and AhlC (green), or pre-incubated with 500 nM AhlA (black), AhlC (blue), AhlB and AhlC (purple), or AhlA and AhlC (orange), for 1 h before addition of the remaining proteins at time 0. **d** % lysis against time when erythrocytes were incubated with 500 nM AhlB and AhlC (green), or pre-incubated with 500 nM AhlC (blue) or AhlB (purple) for 1 hour before addition of the second protein at time 0. **e** TEM negative stain images of AhlB, AhlBC and AhlABC pores in both erythrocytes and liposomes. Scale bars on magnified images represent 10 nm, and pores are highlighted in red circles. **f** Liposome floatation assay to determine lipid bilayer binding ability of each Ahl toxin component. Membrane binding ability of each AhlA, AhlB and AhlC alone and AhlB+AhlC, and AhlA+AhlB+AhlC (+) was assessed by SDS-PAGE analysis of the six top and six bottom fractions (left to right gel lanes with molecular weight indicated, kDa), together with a control without liposomes (−). A schematic of the ultracentrifuge tube shows the location of the top and bottom fractions and in which fractions liposomes and protein are expected. TEM negative stain EM images are shown of top fractions of AhlB (top), AhlBC (centre) and AhlABC (bottom), with pores circled in red. **g** SDS-PAGE of soluble (sol) and pelleted (pel) fractions from ultracentrifugation of AhlA, AhlB and AhlC with detergent, alone and in combination. Invitrogen Mark12 ladder (L) is labelled in the first lane. All assays were carried out in triplicate ($n = 3$) with error bars corresponding to the mean ± standard deviation. Source data are provided as a Source Data file

## Table 1 X-ray data collection and refinement statistics for AhlB structures

|  | AhlB soluble PDB: 6GRK | AhlB pore SeMet PDB: 6GRJ | AhlB pore Form 2 PDB: 6H2F |
|---|---|---|---|
| *Data collection* |  |  |  |
| Beamline | I03 | I03 | I02 |
| Wavelength (Å) | 0.9794 | 0.9794 | 0.9795 |
| Space group | C2 | C2 | C222₁ |
| *Cell dimensions* |  |  |  |
| a, b, c (Å) = | 133.5, 79.8, 111.0 | 363.6, 116.5, 217.4 | 117.7, 178.2, 485.6 |
| α, β, γ (°) = | 90, 90.2, 90 | 90, 118.0, 90 | 90, 90, 90 |
| Molecules per asymmetric unit | 3 | 10 | 10 |
| Resolution (Å)[a] | 2.33–58.35 (2.33–2.39) | 2.94–107.58 (2.94–2.99) | 2.55–39.55 (2.55–2.73) |
| Total reflections[a] | 162245 (12843) | 1150390 (52927) | 1053972 (195594) |
| Unique reflections[a] | 49042 (3660) | 169188 (8317) | 165659 (29762) |
| $R_{merge}$[a,b] | 0.074 (0.50) | 0.20 (1.8) | 0.19 (0.63) |
| $R_{pim}$[a,c] | 0.070 (0.46) | 0.082 (0.76) | 0.12 (0.40) |
| Mean I/σ(I)[a] | 9.1 (2.0) | 6.7 (1.0) | 6.6 (2.7) |
| Completeness (%)[a] | 98.1 (99.8) | 99.1 (98.0) | 100 (100) |
| Multiplicity[a] | 3.3 (3.5) | 6.8 (6.4) | 6.4 (6.6) |
| Mid-slope |  | 1.025 |  |
| dF/F |  | 0.186 |  |
| *Refinement* |  |  |  |
| No. of non-H atoms | 7875 | 24386 | 24766 |
| Rwork/Rfree | 0.22/0.25 | 0.22/0.24 | 0.19/0.28 |
| Average B factors (Å²) | 49 | 70 | 42 |
| Bond length rmsd (Å) | 0.0095 | 0.011 | 0.011 |
| Bond angle rmsd (°) | 1.41 | 1.57 | 1.57 |
| Ramachandran favoured/allowed (%) | 95.8/100 | 97.9/100 | 96.1/100 |

[a]Values in brackets are for data in the high-resolution shell
[b]$R_{merge} = \Sigma_{hkl} \Sigma_i |I_i - I_m| / \Sigma_{hkl} \Sigma_i I_i$
[c]$R_{pim} = \Sigma_{hkl} \sqrt{1/n - 1} \Sigma_{i=1} |I_i - I_m| / \Sigma_{hkl} \Sigma_i I_i$, where $I_i$ and $I_m$ are the observed intensity and mean intensity of related reflections, respectively

**Structure of the pore form of AhlB**. We determined the structure of an oligomeric pore conformation of AhlB, by growing crystals in the presence of MPD, a reagent which has been previously shown to induce pore formation in PFTs[34]. The structure of AhlB was determined to 2.9 Å in space group C2 (Form 1), using Se-Methionine SAD and subsequently the resolution was extended to 2.5 Å, using data from a crystal in space group C222₁ (Form 2, Table 1, Supplementary Fig. 4). The structures of both crystal forms are closely related (rmsd = 0.78 Å), with the asymmetric unit in each case containing ten copies of AhlB arranged as a ring of subunits in C5 symmetry, forming a funnel shape around a central pore, with an overall length of 143 Å and an external diameter of 115 Å at the large end of the funnel (the tail), reducing to a minimum diameter of 30 Å at the neck before finally expanding to a final diameter of 46 Å (the head), dimensions in agreement with those observed in negative stain EM for AhlB pores (Fig. 3a). The internal dimensions of the pore were calculated using HOLE[35], which showed a minimal internal diameter at the neck of 20 Å (Fig. 3b).

Each subunit of the pore is constructed from two extended α-helices (α3 and α4) that join the head of the pore to the five α-helices that fold into the pore tail. Major conformational changes have occurred when compared to the soluble structure, which are centred on rotations about two hinges, hinge 1 (K152-L160 and R248-A254) and hinge 2 (L186-G191 and A222-G225; Fig. 4). In this soluble to pore transition, changes in ϕ and ψ angles of up to 180° for the hinge residues and movements of up 94 Å result in both the β-tongue and α3a/α4a (D156-L186 and G225-Q246) unpacking and the secondary structure rearranging to form the two 140 Å extended helices (α3 and α4), constructing the head of the funnel and presenting this hydrophobic head to the membrane (Fig. 4 and Supplementary Movie 1). The stretch of adjacent hydrophobic head residues at the C-terminus of α3 (G176–A201) and the N-terminus of α4 (V212–L234), each of length 39 Å is sufficient for these two helical segments to insert fully through both leaflets of a lipid bilayer and are highly conserved in all Gram negative bacteria (Supplementary Fig. 5), with the extent of the insertion of the helices into the membrane defined by rings of tyrosine residues on α4 (Y245) and phenylalanine residues on α3 (F203; Fig. 3c), residues that are known to delineate transmembrane helices[36]. In the two crystal forms of the AhlB pore, the helices of the head domain adopt slightly different arrangements at their distal ends, due to differences in crystal packing between the two structures (Supplementary Fig. 6), indicating that there is some flexibility in the packing in the transmembrane part of the pore.

Within the oligomeric pore, AhlB adopts two subunit conformations, which vary at both the tail and head regions. In the Type 2 conformation residues from both hinges adopt helical

structure of AhlB all non-Ala/Gly hydrophobic residues of the hydrophobic head are packed away in the core of the protein as the hydrophobic β-tongue. The use of such a β-tongue to shield membrane inserting hydrophobic residues is a characteristic of the soluble conformations of many ClyA family members, including Hbl-B, ClyA, MakA and NheA[8,23,24,33].

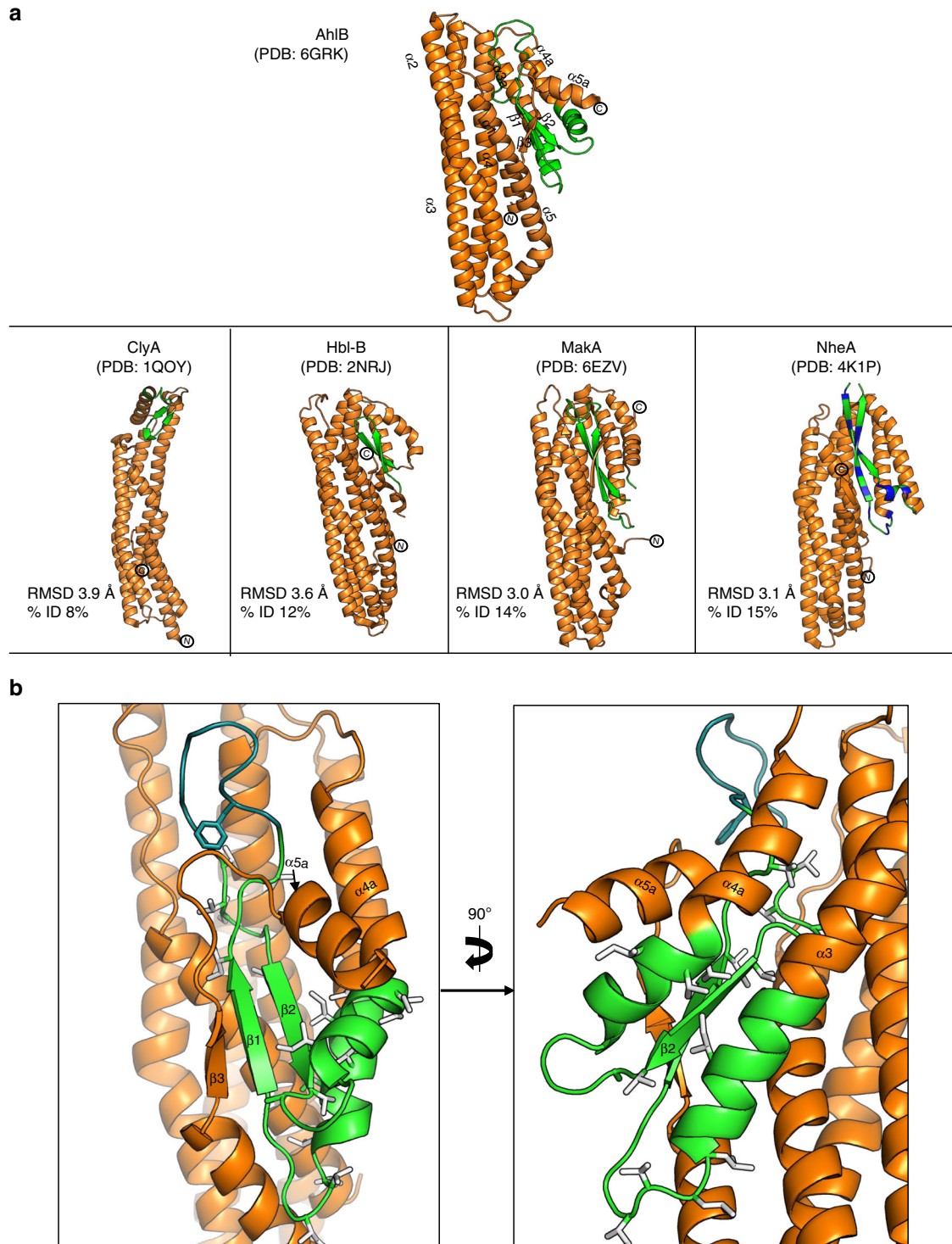

**Fig. 2** β-tongue structure of soluble AhlB. **a** Structure of Soluble AhlB and comparison with other related structures. The region of the head that forms the two membrane spanning helices in the pre-pore form of AhlB are highlighted in green, below are soluble structures of ClyA, Hbl-B, MakA and NheA, the proposed hydrophobic regions coloured green with the exception of NheA where the amphipathic β-tongue is coloured green (hydrophobic) and blue (hydrophilic), RMSD and % sequence Identity to soluble AhlB are given. **b** Hydrophobic β-tongue of soluble AhlB, coloured as in (**a**). All non-Ala/Gly hydrophobic residues of the head are highlighted as sticks

conformations, forming sections of α3 and α4, whereas in the Type 1 conformation, the hinge 1 residues in α3 (K152–L160) remain in a loop, indicating that the Type 2 conformation is the fully extended form. Indeed, the transition to form AhlB Type 2 from AhlB Type I is an extension of the same hinge movement observed between the soluble and type I conformations

(Supplementary Movie 1). Furthermore, in Type 1 the N-terminal helix (α1) is sandwiched between helices α4 and α5, whereas in the Type 2 conformation the first 15 residues of α1 are disordered, with the N-terminus of α1 packing against the C terminus of α5 in an end-to-end arrangement (Fig. 3c). Thus, the Type 1 conformation has a five-helix bundle at the tail, whereas

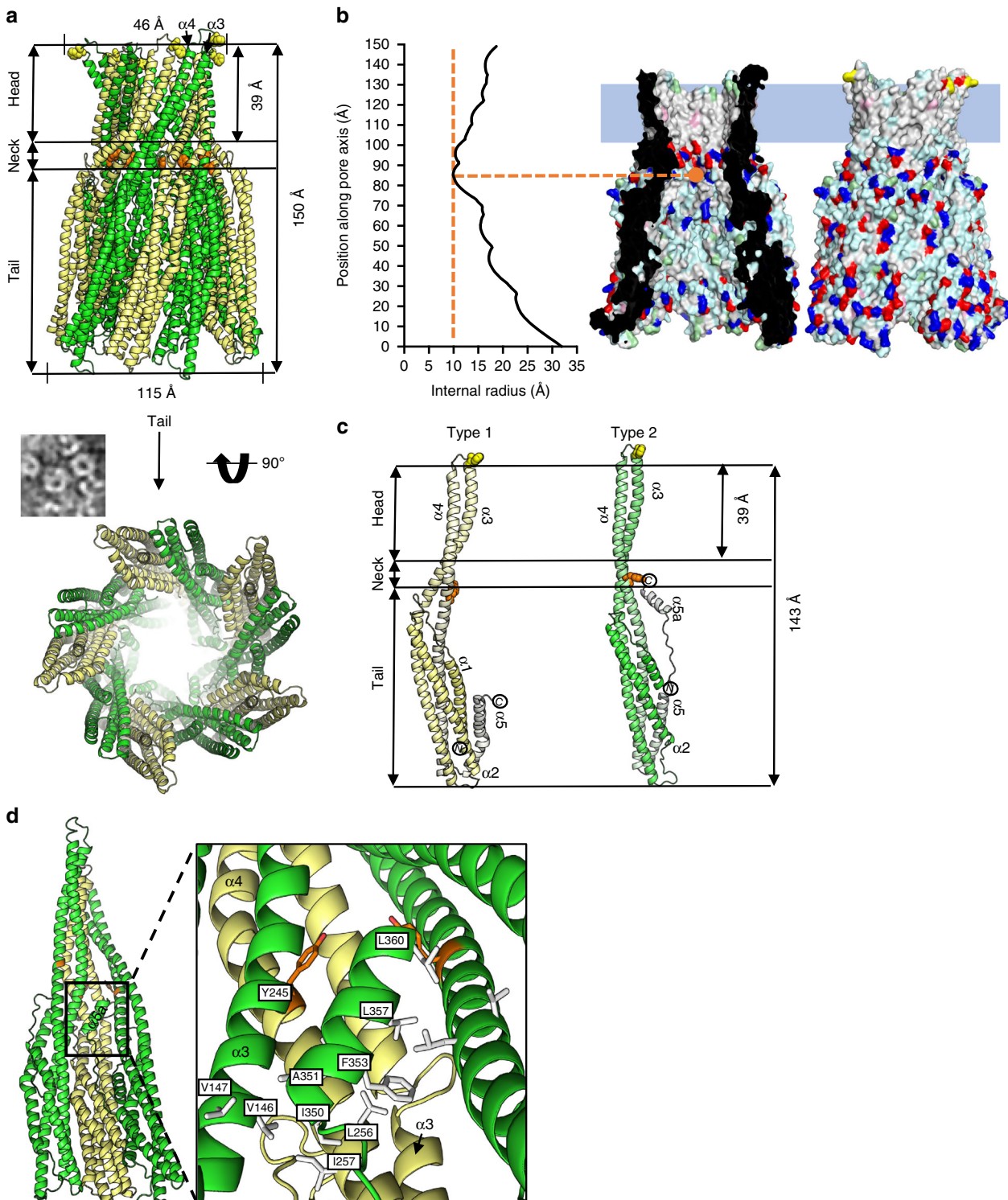

**Fig. 3** Structure of the AhlB pore. **a** Cartoon representation of AhlB pore, Type1 conformation is pale yellow while Type 2 is coloured green. Y245 is shown as orange spheres and delineates the beginning of the hydrophobic head, with the ring of F203 highlighted as yellow spheres at the end of the head. Below is a negative stain EM image of AhlB pores in liposomes next to the crystal structure, orientated as seen in the EM, with the a view looking down the pore from the tail. **b** Surface rendering generated in PyMOL[61], shows hydrophobic (white), negative (red), positive (blue) and polar (cyan) surfaces. The membrane bound head domain is entirely hydrophobic, with a blue rectangle defining the proposed membrane region. The internal radius against position along the vertical pore axis is plotted with the narrowest region in the neck labelled, calculations were performed using HOLE[35]. **c** Side view of the AhlB Type 1 and Type 2 protomers. Protomers are coloured dark (N-terminus) to light (C-terminus). The Head, neck and tail are highlighted along with residues F203 and Y245 coloured as in (**a**). **d** Interactions between α5a with the neighbouring subunits in the Type 1/Type 2 dimer, with hydrophobic residues highlighted in white and Y245 coloured orange

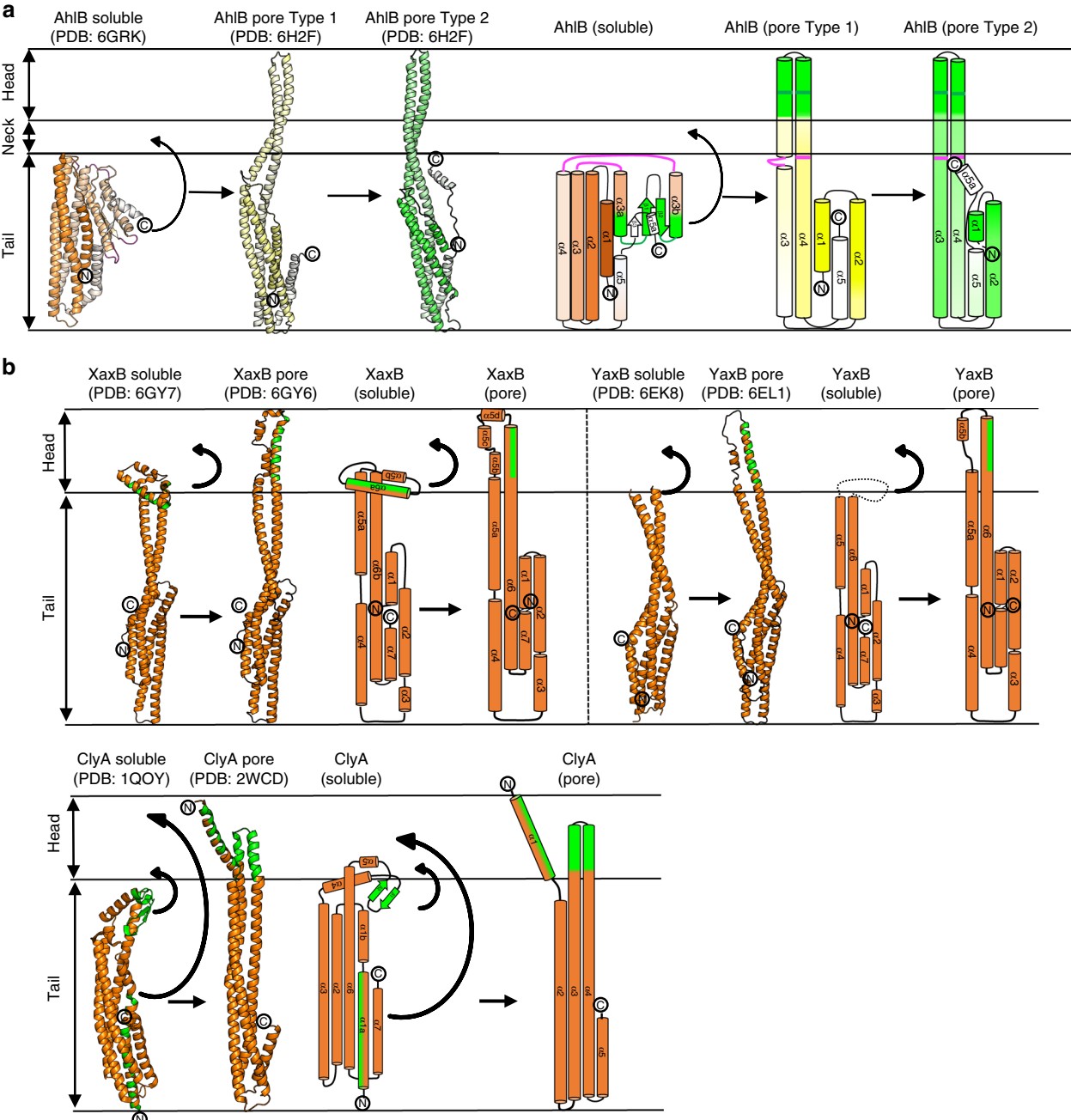

**Fig. 4** Differences in conformation between soluble and pore form AhlB. The β-tongue of soluble AhlB undergoes two 180° rotations into the AhlB pore Type 1 structure to expose its hydrophobic head, followed by untwisting of the head and movement of α1 and α5 to the Type 2 structure. **a** Cartoon structures of soluble AhlB (orange) and pore form AhlB Type 1 (yellow) and AhlB Type 2 (green), coloured from dark (N-terminus) to light (C-terminus). Schematic diagrams of AhlB coloured according to (**a**), with the addition of hydrophobic regions coloured in green. The location of the two hinge regions in the soluble AhlB structure are marked on the pore structures as pink and green lines. **b** Cartoon (left) and schematic (right) diagrams of XaxB, YaxB and ClyA soluble and pore forms. The β-tongue of soluble ClyA undergoes two rotations similar to that of AhlB to form a short hydrophobic head, long enough to span a single leaflet of a lipid bilayer, while the N-terminus swings out to form an amphipathic membrane spanning helix. Unlike ClyA and AhlB, YaxB and XaxB have no β-tongue and a single hinge, which unfolds to expose a single amphipathic helix

the Type 2 conformation has a four-helix bundle at the tail. The packing arrangement of the N and C terminal helices in the tail of Type 1 is the same as that observed in soluble AhlB, as well as soluble ClyA, NheA, Hbl-B and MakA[8,23,24,33] (Supplementary Fig. 7). In contrast, the Type 2 conformation in this tail region packs into an arrangement similar to that seen in the pore structures of both YaxB and XaxB[11,15] (Supplementary Fig. 7), suggesting that the fully extended Type 2 conformation of AhlB is likely to be the conformation present in the active AhlABC pore.

In the AhlB pore one of each of the subunit conformations of AhlB pack together to form a dimer and five of these Type 1/Type 2 dimers assemble into the decameric oligomer (Fig. 3). The interface between dimers is constructed from α5a of the Type 2 AhlB monomer (residues I350, A251, F353, L357 and L360) packing against α4 of the Type 1 AhlB monomer (residues L256 and I257) and α3 in AhlB Type 2 (residues V146 and V147) of the neighbouring dimer. The first 15 N-terminal residues of the Type 2 subunits point towards the centre of the oligomer where

discontinuous electron density is observed (Supplementary Fig. 8), perhaps suggesting that these residues may form a plug in the entrance to the funnel resulting in an inactive pore, possibly explaining the lack of haemolytic activity of AhlB alone. We note, however, that the equivalent residues in the structures of the active YaxAB and XaxAB pores are also disordered[11,15].

**Crystal structure of AhlC soluble tetramer.** As we have shown that the binary complex of AhlB and AhlC is partially haemolytic, the structure of the soluble form of AhlC was determined to 2.8 Å resolution in space group P6₅22 using Se-Methionine SAD, and refined to higher resolution (2.35 Å) using native data (Table 2, Supplementary Fig. 9). The asymmetric unit contained a dimer of AhlC (chains P and Q), which by rotation about the crystallographic twofold axis formed a 222 tetramer (subunits PP', QQ') (Fig. 5), consistent with gel filtration results where AhlC eluted as a tetramer (Supplementary Fig. 2).

Each subunit of AhlC had a very similar structure to that of the Type 2 conformation of the AhlB pore structure, (rmsd 2.4 Å, Fig. 6a), with a bundle of five α-helices forming the tail and helices α3 and α4 running the entire length of the protein and extending from the tail by 50 Å to form the neck and head domains (Fig. 5a). A stretch of 16 residues (L155-V171) at the end of α3 and the start of α4 in the head domain are hydrophobic, clearly indicating a possible method of membrane attachment. However, these sections of the two helices only extend over 15 Å, sufficient to span just one leaflet of the membrane, unlike AhlB where the hydrophobic parts of the equivalent structural elements can span the whole bilayer. The extent of this hydrophobic head domain is marked by a group of conserved lysine residues (K150, K151 and K152) and a tyrosine (Y154) on α3, which would delineate the extent of membrane insertion (Fig. 5b).

Within the tetrameric assembly of soluble AhlC residues L156, L158, L160 and L163 in the α3–α4 hydrophobic head of subunit P pack against the hydrophobic residues L246 and L249 in the tails of both subunit Q and the symmetry related subunit P', to form leucine zippers in the head to tail arrangement of the tetramer. Y154 of subunit P also forms π-stacking interactions with F250 in P', and K152 of subunit P forms an ionic interaction with E257 of chain Q (Fig. 5b). The head of subunit P' adopts the equivalent interactions in the tetramer. The other two hydrophobic heads from subunits Q and Q' are more exposed on the tetramer surface and in the crystal pack against neighbouring hydrophobic heads from symmetry related tetramers (Supplementary Fig. 10). We have also determined the structure of a second crystal form of soluble AhlC in space group P2₁ (Table 2, Supplementary Fig. 9), which contains a complete tetramer in the asymmetric unit. In this second crystal form the head domains of subunit Q and S do not form crystal contacts and the electron density for these areas is poor, indicating that these hydrophobic head domains are flexible within the tetramer (Supplementary Fig. 10). Thus, AhlC appears to use assembly of the quaternary structure to hide its hydrophobic head from the cytosol, rather than the conformational change mechanism seen in AhlB. Alignment of AhlC with YaxA or XaxA in their respective pores show that AhlC is closely related to both these structures, but that the extended conformation of AhlC subunit Q is more similar to YaxA or XaxA than

**Table 2 X-ray data collection and refinement statistics for AhlC structures**

| | AhlC Form 1 (SeMet) | AhlC Form 1 PDB: 6H2E | AhlC Form 2 PDB: 6H2D | AhlC head mutant PDB: 6R1J |
|---|---|---|---|---|
| *Data collection* | | | | |
| Beamline | I04 | I04 | I03 | I04 |
| Wavelength (Å) | 0.9763 | 0.9763 | 0.9763 | 0.97951 |
| Space group | P6₅22 | P6₅22 | P2₁ | P6₁22 |
| Cell parameters | | | | |
| a, b, c (Å) = | 134.6, 134.6, 145.3 | 134.7, 134.7, 145.3 | 65.1, 61.7, 130.0 | 88.5, 88.5, 291.0 |
| α, β, γ (°) = | 90, 90, 120 | 90, 90, 120 | 90, 92, 90 | 90, 90, 120 |
| Molecules per asymmetric unit | 2 | 2 | 4 | 2 |
| Resolution (Å)[a] | 2.81-29.54 | 2.35-58.33 | 2.62-46.74 | 1.92-76.68 |
| | (2.81-2.88) | (2.35-2.41) | (2.62-2.69) | (1.92-1.95) |
| Total reflections[a] | 751,732 | 1,293,084 | 105,013 | 988,355 |
| | (51,617) | (97,395) | (7858) | (22,848) |
| Unique reflections[a] | 19,468 | 32,942 | 31,150 | 52,765 |
| | (1369) | (2392) | (2309) | (2550) |
| $R_{merge}$[a,b] | 0.129 | 0.138 | 0.074 | 0.084 |
| | (1.013) | (3.967) | (0.710) | (2.073) |
| $R_{pim}$[a,c] | 0.029 | 0.023 | 0.058 | 0.019 |
| | (0.230) | (0.639) | (0.509) | (0.710) |
| Mean $I/\sigma(I)$[a] | 28.1 (4.8) | 19.8 (1.4) | 11.0 (1.8) | 16.1 (0.9) |
| Completeness (%)[a] | 99.8 (98.1) | 99.8 (99.8) | 99.7 (100) | 100 (100) |
| Multiplicity[a] | 38.6 (37.7) | 39.3 (40.7) | 3.4 (3.4) | 18.7 (9.0) |
| Mid-slope | 1.36 | | | |
| dF/F | 0.059 | | | |
| *Refinement* | | | | |
| No. of non-H atoms | | 3866 | 6152 | 4108 |
| $R_{work}/R_{free}$ | | 0.22/0.27 | 0.27/0.33 | 0.23/0.28 |
| Average B factors (Å²) | | 68 | 58 | 51 |
| Bond length rmsd (Å) | | 0.012 | 0.011 | 0.0092 |
| Bond angle rmsd (°) | | 1.50 | 1.48 | 1.61 |
| Ramachandran favoured/allowed (%) | | 96.4/100 | 93.7/100 | 99.0/100 |

[a]Values in brackets are for data in the high-resolution shell
[b]$R_{merg} = \Sigma_{hkl} \Sigma_i |I_i - I_m| / \Sigma_{hkl} \Sigma_i I_i$
[c]$R_{pim} = \Sigma_{hkl} \sqrt{1/n - 1} \Sigma_{i=1} |I_i - I_m| / \Sigma_{hkl} \Sigma_i I_i$, where $I_i$ and $I_m$ are the observed intensity and mean intensity of related reflections, respectively

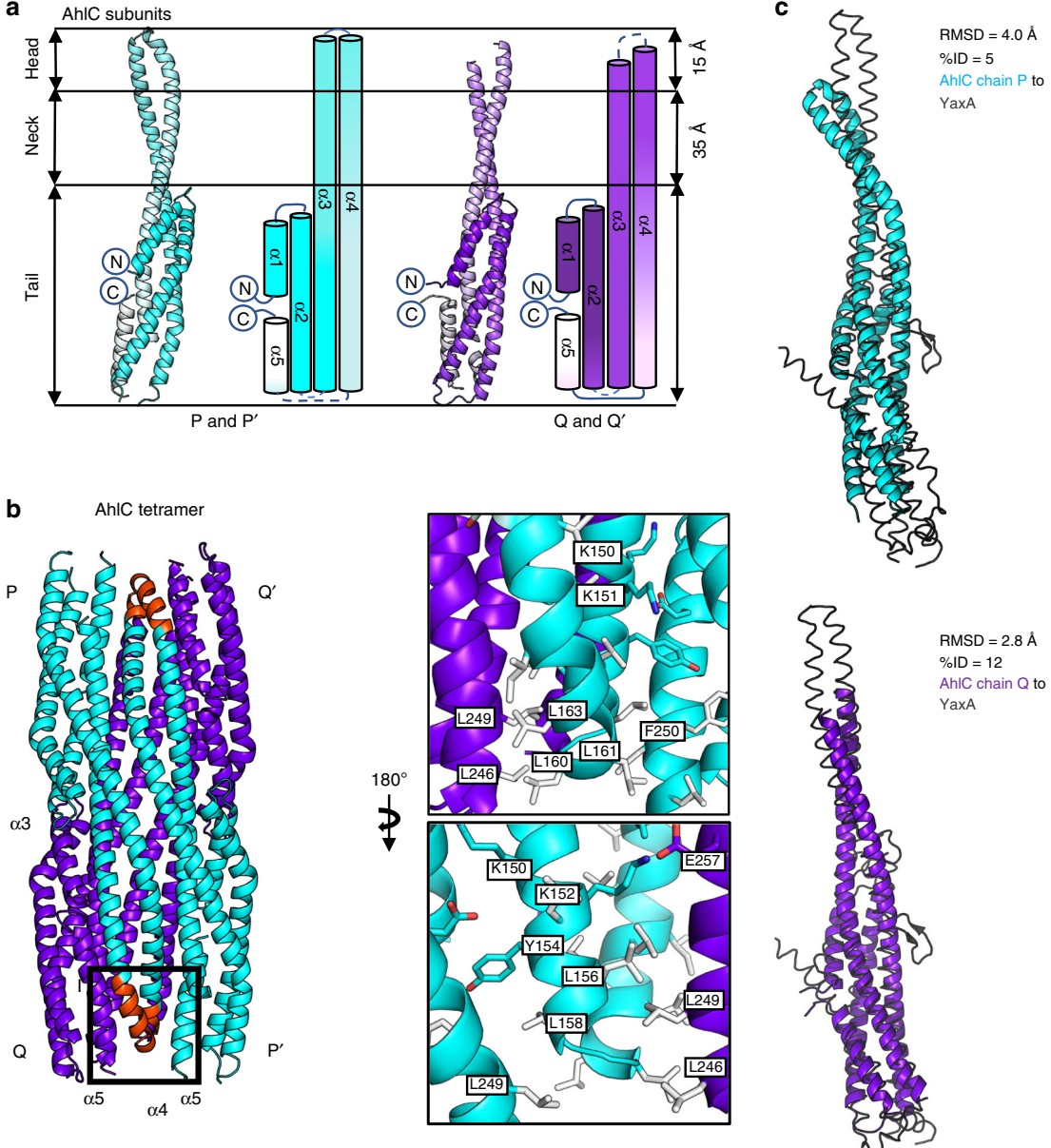

**Fig. 5** The AhlC soluble tetramer buries its hydrophobic heads. **a** Cartoon and schematic structures of AhlC Form 1 subunits P and P' (cyan) and Q and Q' (purple), coloured dark (N-terminus) to light (C-terminus). **b** Tetrameric assembly of AhlC, hydrophobic heads of α3 and α4 are highlighted (red) in subunit P and form leucine zipper interactions with α5 in the tails of both subunit Q and subunit P'. Hydrophobic residues are coloured white. **c** Superposition of YaxA (grey) with the two independent subunits of subunit P and Q of the AhlC tetramer, as aligned by the DALI server [59], showing higher structural similarity to subunit Q

that of AhlC subunit P, where the head is buried within the tetramer (rmsd 2.8 and 2.7 Å; and 4.0 and 2.8 Å, for subunits Q and P against YaxA and XaxA, respectively, Fig. 5c).

To determine if disassembly of the tetramer is required for membrane binding, we cross-linked soluble AhlC with glutar-aldehyde to produce equal proportions of AhlC monomer and AhlC dimer, whereas when AhlC was cross-linked with both glutaraldehyde and 1-ethyl-3-(3-dimethylaminopropyl)carbodii-mide, the dimeric species was formed exclusively (Supplementary Fig. 11a). Haemolytic assays using these cross-linked samples showed that lysis was abolished for the purely dimeric form and reduced by about 50% for the mixture of monomeric and dimeric AhlC. Based on the length of the hydrophobic head of AhlC, its delineating charged residues and the observation that it is not lytic in isolation, we suggest that AhlC inserts into a single leaflet

of the bilayer. We thus constructed a triple leucine to threonine mutation (α3 L156T, and α4 L160T, L161T) of AhlC to reduce the hydrophobicity of the head. Haemolytic assays using the triple mutant of AhlC resulted in an almost 80% reduction in activity of the AhlABC complex (Supplementary Figure 11b). We also determined the structure of this triple mutant of AhlC (Table 2), which showed it adopts the same tetrameric structure as that observed for the wild-type AhlC (rmsd 0.6 Å) . Taken together, these results show that monomeric AhlC is required for lysis, and that disassembly of the tetramer of AhlC and the hydrophobic nature of the head of AhlC are both vital for pore formation.

## Discussion

Within the greater ClyA family there are a number of require-ments that need to be met in constructing a fully active pore

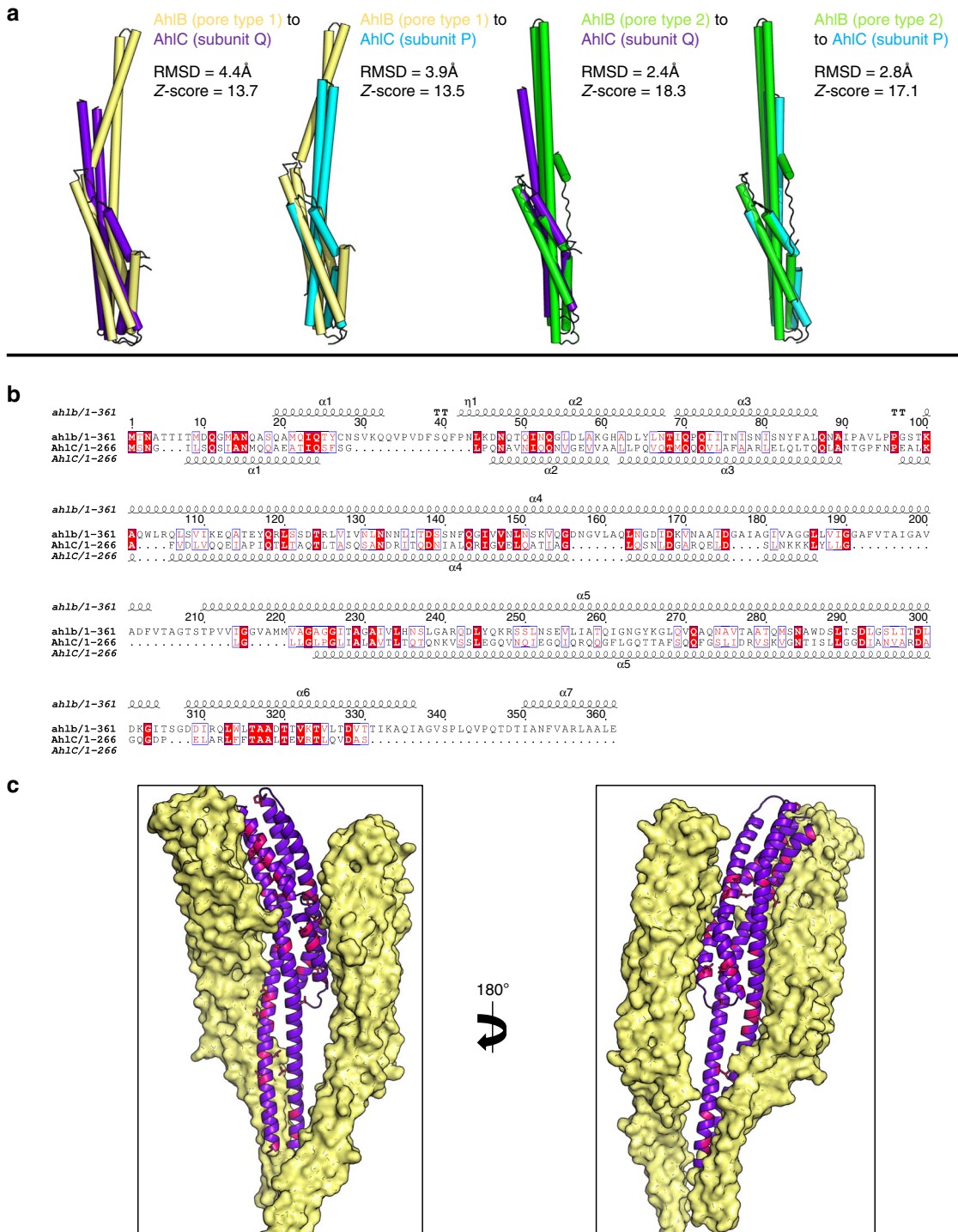

**Fig. 6** Conserved residues between AhlB and AhlC. **a** Superposition of AhlB Type1 (yellow) and Type 2 (green) with subunits P and Q of the AhlC tetramer (blue and purple respectively), as aligned by the DALI server[59], showing higher structural similarity to subunit Q. **b** Sequence alignment of AhlB and AhlC. Identical residues are highlighted in red boxes, with similar residues in red lettering. Secondary structure is shown above (AhlB) and below (AhlC). **c** One AhlB Type 2 subunit is replaced by a single AhlC subunit Q (purple) in the AhlB pore. Conserved interface residues on AhlC are coloured pink, with AhlB Type 1 coloured pale yellow

forming system. These include masking the hydrophobic membrane attachment residues from solvent in the soluble form of the protein; the transition from soluble to pore form; the initial attachment to the membrane and the oligomerization to form the active pore. For the simplest member of the family, the *E. coli* protein ClyA itself, these four functions are carried on the same polypeptide. The ClyA protomer unpacks from a compact soluble form, refolding its β-tongue region into the hydrophobic α3–α4 head that inserts into one leaflet of the membrane, whilst further conformational changes expose the amphipathic N-terminal helix which penetrates both membrane leaflets and forms the lining of the pore upon oligomerization[9,10].

Comparison of the soluble and pore structures of AhlB shows that a major rearrangement of the protein occurs between them,

in a similar way to that seen in the ClyA pore formation. The hydrophobic β-tongue of AhlB is shielded from solvent by packing against α3a, α4a, the N-terminal helix and C-terminal helix of the tail domain in the soluble conformation. A bifold hinge mechanism operates to unpack the β-tongue from the tail domain to form the 140 Å extended α3 and α4 helices, refolding the hydrophobic β-tongue into a α-helical conformation to provide the membrane spanning α3–α4 head domain, in ClyA these conformational changes have been proposed to occur via a molten globule state[10]. Unlike in ClyA where the α3 and α4 head inserts into just one leaflet[9], the equivalent hydrophobic head region of AhlB is of sufficient length to cross both membrane leaflets, indicating that AhlB may be the pore forming component of the system. A similar role is played by both YaxB and XaxB in their respective bipartite pores, where the α5–α6 head (equivalent to α3–α4 in AhlB) inserts through both leaflets of the membrane, with a similar, but smaller conformational change to that seen for AhlB or ClyA[10,11,15] (Fig. 4). The α5–α6 head of both YaxB and XaxB protomers is formed from amphipathic helices making a hydrophilic pore lining, whereas both the head in AhlB, and thus the pore lining, are hydrophobic.

The attachment of a toxin component to a single leaflet of the target bilayer occurs in ClyA, and for the YaxA and XaxA components of the YaxAB and XaxAB bipartite PFTs[10,11,15]. The extent of hydrophobic residues in the α3–α4 head of AhlC is only sufficient to insert into one leaflet and thus AhlC most likely carries this role in the AhlABC tripartite PFT, supported by liposome float assays and associated negative stain EM which show that AhlC associates with a bilayer, but that no pores are formed from AhlC alone. In addition, the triple Leu–Thr head mutant of AhlC designed to decrease its hydrophobicity, reduces lytic activity by 80%, emphasising the head's critical role. We suggest the single leaflet insertion of AhlC is the initial binding event, as the lag time for lysis is abolished when erythrocytes are pre-incubated with AhlC, a situation also observed for the Nhe and Hbl tripartite PFTs of *B. cereus* where NheC and Hbl-B (each equivalent to AhlC) have been shown to attach to membrane lipids to start pore assembly[37,38].

For AhlC membrane attachment the hydrophobic head must become exposed. In ClyA this is achieved during the conformational change from the soluble to pore form[9,10]. However, the AhC subunits of the soluble tetramer adopt a pore type conformation with the hydrophobic heads buried within the tail domains of symmetry related subunits. Thus some disassembly of the AhlC tetramer must occur before membrane attachment, supported by the cross-linking of AhlC, which abolishes lytic activity. Flexibility of the head to disassociate from the tetramer when presented with a hydrophobic surface is also shown in the crystals of AhlC, where the α3–α4 hydrophobic heads of two subunits protrude from the tetramer to pack against equivalent regions of symmetry related molecules in one crystal form and make no contacts and are disordered in the other. Such disassembly of the AhlC tetramer on binding to membrane would produce a distinct concentrating effect of initiating subunits on the membrane surface, explaining the rapid lysis of erythrocytes pre-incubated with AhlC before addition of the other pore components. As the structure of AhlC is very similar to the AhlB Type 2 conformation, with 30 out of the 55 conserved residues between AhlB and AhlC present at the AhlB Type1/Type 2 interface (Fig. 6) a further role for AhlC may be to instigate the conformational change in AhlB by providing a similar structure and binding surface to that seen between protomers in the AhlB pore.

Multiple sequence alignment of the C components of the α-PFTs that we have identified from other Gram-negative bacteria show that residues involved in the packing interactions that conceal the hydrophobic head in the AhlC tetramer are conserved (Supplementary Fig. 12). Residues K152 and E257 which form a salt bridge between the head and tail domains in AhlC are identical in all the proteins with the exception of *Moritella* sp., *Janthinobacterium lividum* and *Serratia* sp. where conservative substitutions occur (K152Y and E257S, respectively). The hydrophobic heads of all the C components are 15 residues long and are rich in Leu and Ile with L158 in α3, L163 in α4 and L249 in α5 conserved and could thus form the same leucine zippers as seen in AhlC. This suggests that the tetrameric assembly of soluble AhlC is conserved across these tripartite α-PFTs of Gram-negative bacteria (Supplementary Fig. 12). It thus appears that in the tripartite AhlABC system the initial transition from soluble to membrane forms of AhlC and binding into a single leaflet occurs via a different mechanism to ClyA, based on the disassembly of a tetramer rather than the conformation change mechanism seen in ClyA[9,10]. In contrast, the hydrophobic residues of YaxA and XaxA (equivalent to AhlC) are not concealed in their respective monomeric soluble structures, although these residues do form hydrophobic interactions in their crystal lattices, perhaps suggesting that soluble YaxA and XaxA may fold in a manner similar to ClyA or AhlB to hide their hydrophobic residues.

The structures of the tripartite Ahl PFT proteins AhlC and the pore form of AhlB presented here are closely related to those of the bipartite PFTs YaxAB and XaxAB (Supplementary Figs. 7 and 13). The decameric AhlB pore is constructed from five dimers of the Type1 and Type 2 conformations, with the Type 1 conformation an intermediate on the soluble to pore transition (Supplementary Movie 1). As we have shown that AhlBC pores are partially active, and as an equal number of the single leaflet insertion heads surround the membrane spanning pore in both the ClyA and YaxAB/XaxAB pores, it would thus seem reasonable to propose that AhlC and AhlB could assemble into a pore containing 10 copies of AhlB and 10 of AhlC, with the ring of AhlB subunits surrounded by a ring of AhlC subunits (AhlC is equivalent to YaxA). However, as 80% lysis can also be achieved with a ratio of AhlC:AhlB:AhlA of 0.25:1:1 (Fig. 1b), pores with fewer copies of AhlC might also be lytic. Nevertheless, using the YaxAB pore as a guide, we modelled AhlB and AhlC into a decameric pore. Using alternating AhlB Type 1 and Type 2 conformations with AhlC produced a pore with severe steric clashes at the head. However, if a pore is made using the AhlB type 2 conformation alone with AhlC, no steric clashes occur (Supplementary Fig. 14), further indicating that the AhlB Type 2 conformation is likely to be that of the active pore structure.

This model of an AhlBC pore has a constriction diameter of approximately 30 Å, slightly larger than that seen in the AhlB pore structure (20 Å), but similar in size to both the YaxAB and XaxAB structures (31 Å)[11,15] and consistent with the observation that molecules larger than ~20 Å cannot pass through a similar NheBC pore from the tripartite *B. cereus* Nhe system[22]. However, this proposed AhlBC pore has a hydrophobic pore lining, as the α3–α4 head of AhlB is formed exclusively from hydrophobic residues. This is in direct contrast to ClyA, XaxAB and YaxAB, where the pore lining itself is hydrophilic. As maximal lysis only occurs when all three components of the AhlABC system are present, AhlA must play an important role in the pore construction. Indeed, the liposome float assays and complementary ultracentrifugation experiments with detergent show that all three components are present in pores formed from AhlA, AhlB and AhlC and EM images of liposome preparations show that efficient pore formation, with pores visible in every liposome, only occurs when AhlA, AhlB and AhlC are incubated together with the membranes, compared to the situation with AhlB/AhlC alone, when pores can be seen in just some of the liposomes (Fig. 1e and Supplementary Fig. 3e).

One possible function of AhlA is that it provides the hydrophilic pore lining. AhlA shares closest sequence similarity with *B. cereus* NheA (39%) enabling a structure based sequence alignment to be constructed between NheA (PDB code 4k1p) and AhlA (Supplementary Fig. 15). In this alignment, β1 of NheA (N196–T214) is equivalent to residues D189–T207 of AhlA, with residues K230-S248 in AhlA equivalent to β2 of NheA (T217–A235). There is an insertion in the AhlA sequence between β1 and β2, compared to NheA, which is also seen in the other A components identified in Gram-negative bacteria. Assuming a similar bifold soluble to pore transition for AlhA as seen in AhlB, then these β1–β2 residues would form the extended membrane spanning helices (α3–α4) and would be of sufficient length to pass through both leaflets of the membrane, with S208–A229 in AhlA forming a intracellular loop between them. The sequence of these helices shows that α3 of the head would be amphipathic with α4 hydrophilic (Supplementary Fig. 15). These helices could pack together via their hydrophilic surfaces to produce a membrane-spanning element with an overall amphipathic character. It is, therefore, plausible that within the fully active pore, AhlA produces the hydrophilic pore lining with the hydrophobic surface of AhlA α3 packing against the hydrophobic surface of AhlB, and the hydrophilic α4 lining the inside of the pore. As the kinetic assays show that equal amount of AhlA and AhlB are present in the most active pores, the pore could be formed with alternating AhlA and AhlB subunits at the tail and a ring of AhlA α3–α4 heads providing the hydrophilic lining of the membrane-spanning region. AhlA could be accommodated within the pore without reducing the internal diameter by movement of the AhlB heads, which have been shown to be flexible between the two crystal forms of the AhlB pore (Fig. 7). Such an AhlABC pore would necessitate an increased outside diameter for the transmembrane region, compared to the AhlB or AhlBC pores alone, as this pore model would have a splayed funnel shape, like YaxAB and XaxAB, the pore diameter at the funnel entrance would only have to be slightly bigger in the AhlABC pore than the AhlBC pore, in agreement with negative stain EM images that show a similar size for the AhlABC and AhlBC pores. Flexibility in assembling pores is shown in both ClyA and XaxAB PFTs, where pores of variable protomer number

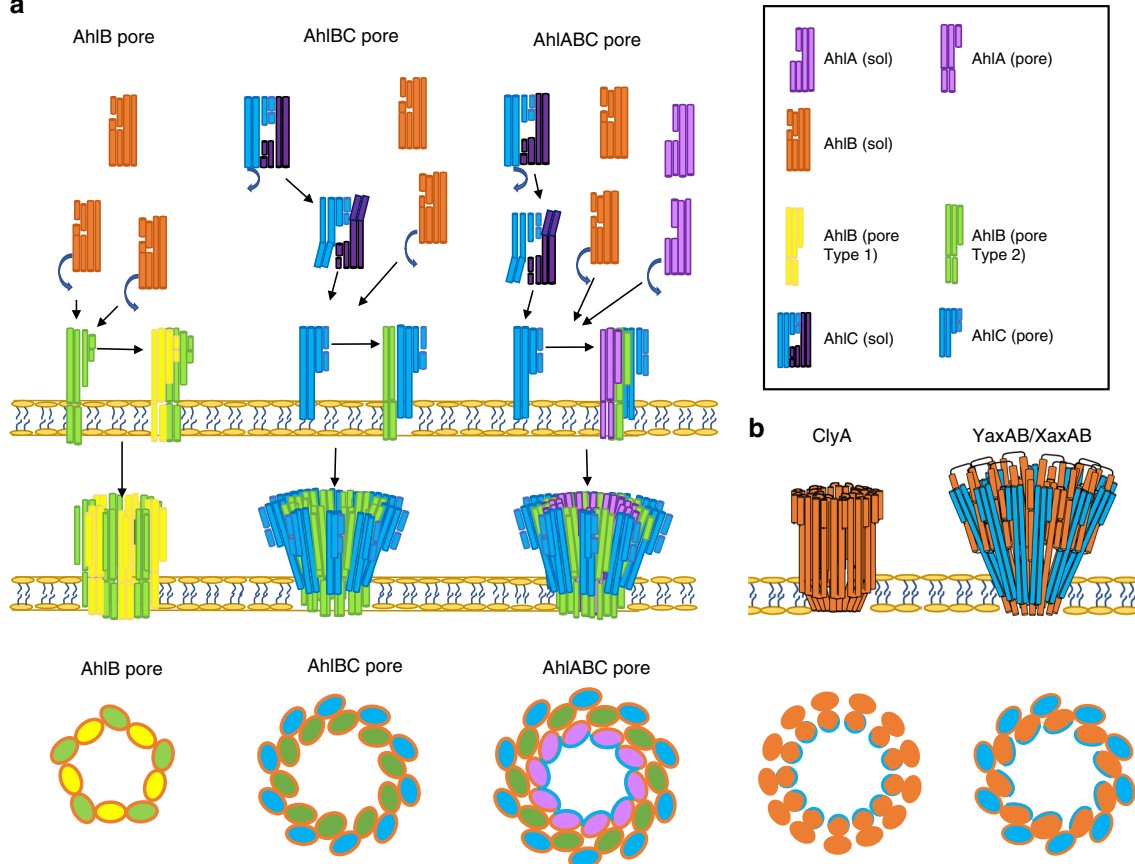

**Fig. 7** A proposed assembly schematic of AhlB, AhlBC and AhlABC pores. **a** Assembly of AhlB pores; soluble AhlB (orange) reconfigures to the Type 1 pore conformation (yellow) on exposure to the lipid bilayer and recruits more AhlB monomers to form an inactive pore of mixed Type 1, Type 2 (green) conformation. Assembly of AhlBC pores; when AhlC (cyan) is present tetramers of AhlC disassemble at the membrane and monomers insert into one leaflet. Soluble AhlB (orange) is recruited to the lipid bilayer where it unpacks (green) to form a hetero dimer with AhlC. Further AhlB and AhlC are recruited until a complete pore is formed with a ring of AhlC on the outside and a hydrophobic ring of AhlB on the inside. Assembly of AhlABC pores; AhlC inserts in to the membrane as in AhlBC pore assembly, soluble AhlA and AhlB are then recruited to the membrane where they associate with AhlC and further AhlA, AhlB and AhlC are recruited until a complete pore is formed with a hydrophilic lining from AhlA on the inside of the membrane spanning region. Shown below is a cross-section views through the membrane bound region of each pore, with AhlB type 1 (yellow), AhlB type 2 (green), AhlC (blue) and AhlA (pink) with hydrophobic surfaces (orange) and hydrophilic (light blue) highlighted. Each oval represents the two α3, α4 head helices. Relative sizes of the proposed pores are not implied by these schematics. **b** Schematics of the ClyA and the YaxAB/XaxAB pores. Below is a cross-section view through the membrane bound region of each pore. ClyA and YaxB/XaxB are coloured orange, while YaxA/XaxaA is coloured blue. Hydrophilic (blue) and hydrophobic (orange) surfaces are highlighted. Both pores have a hydrophilic internal lining

have been observed[15,32], which may also be the case in AhlABC, (for example, with alternating protomers of AhlA and AhlB providing the membrane spanning region), but the overall architecture would be the same. The amphipathic α3 and hydrophilic α4 sequence pattern of AhlA is also conserved within the A component of the other tripartite Gram-negative bacteria α-PFTs we have identified, suggesting that a similar role for the A component may operate throughout the tripartite family (Supplementary Fig. 15c).

The large scale conformational change seen in both AhlB and ClyA and proposed for AhlA presumably also occurs for at least the tripartite PFT components *B. cereus* NheA and HblB, as structures of the soluble forms of these two proteins are in the same conformation as seen in the soluble forms of AhlB and ClyA (Fig. 2). In addition, the *Vibrio cholerae* MakA protein, of the motility associated cytotoxin operon, also shares the same general structure as soluble AhlB, but as yet it is unclear whether the *mak* operon encodes a ClyA family PFT[33]. A similar, but smaller scale conformational change occurs for the bipartite YaxB and XaxB proteins[11,15], indicating that this mechanism occurs across the whole ClyA family.

It is thus clear that initial insertion into a single leaflet by two short hydrophobic α helices is a common feature of all these α-PFT's, carried out by AhlC in the three component AhlABC system, by YaxA or XaxA in the two component YaxAB or XaxAB pores and by the β-tongue in the single component ClyA pore, albeit the method by which the hydrophobic residues are occluded in the soluble forms is different. The membrane spanning pore itself is then assembled using the N-terminal helix of ClyA or the components YaxB or XaxB in the bipartite YaxAB and XaxAB pores. However, it appears that in the tripartite AhlABC pore each component carries a separate role. AhlC provides the initial single leaflet insertion, assembly with AhlB then producing an oligomeric hydrophobic pore, and finally recruitment of AhlA produces a hydrophilic fully active AhlABC pore. Structural and sequence analysis suggests this method of pore assembly may well occur throughout the tripartite α-PFT family.

The structures of the tripartite AhlABC toxin components described here show that each are related with those from the greater ClyA family, although low sequence similarity is seen to the greater family members. The known importance of ClyA, NheABC, HblL1L2B, YaxAB and XaxAB for virulence in their respective organisms[11,17,20,21], suggests that the large number of similar α-PFT systems that we have identified in pathogenic Gram-negative bacteria may also be important for the virulence of these organisms. Disruption of the pore assembly may well thus provide a means to develop new virulence-targeted therapies. We also note that biotechnological applications of protein membrane pores to encapsulate enzymes for biosensors[39] and for nucleotide sequencing[40] are increasing and identification of this greater tripartite α-PFT family may well provide a wider variety of different assemblies to be exploited.

## Methods

**Bioinformatic search.** The amino acid sequence of *B. cereus* NheB was submitted to BlastP[41], which identified the gene AXH33180.1 (*ahlB*) from *A. hydrophila*. Upstream and downstream analysis of the *A. hydrophila* genome identified genes AXH33179.1 (*ahlA*) and AHX33181.1 (*ahlC*). All three genes were aligned with the respective NheABC components using Tcoffee[30] and hydrophobicity plots were generated for each gene using Protscale[42]. The amino acid sequences of AHX33180.1, AXH33179.1 and AHX33181.1 were submitted to BlastP[41] excluding *Aeromonas* species to identify orthologues with *E* values < 0.01.

**Protein cloning and purification.** The ahl genes *ahlA*, *ahlB* and *ahlC* were amplified from genomic DNA of *A. hydrophila* strain AL09–71 using polymerase chain reaction with synthetic oligonucleotide primers (Eurofins) and a Q5 high-

fidelity polymerase (NEB) (Supplementary Table 2). The amplified fragments were subjected to double restriction digest using NdeI/XhoI (NEB) for AhlB, AhlC and AhlC^HM, and NdeI/HindIII (NEB) for AhlA, before ligation by T4 ligase (NEB) into a pET21a expression vector (Novagen). The AhlC head mutant was generated from the cloned AhlC wild-type gene using a Q5 mutagenesis kit (NEB; for primer sequences see Supplementary Table 2). All constructs were designed to contain a C-terminal His$_6$ tag.

Each protein was expressed in an *E. coli* BL21 DE3 expression cell line (NEB). All cultures were grown in LB media at 37 °C, until an OD$_{600}$ of 0.6 was reached, at which point protein expression was induced by addition of 1 mM isopropyl β-D-1-thiogalactopyranoside (IPTG). For AhlA, protein expression was carried out at 16 °C overnight, whilst for AhlB and AhlC protein expression was carried out at 25 °C overnight. Purification of AhlA, AhlB and AhlC was carried out using the same protocol. Cell pellets were resuspended in lysis buffer (50 mM Tris-HCl pH 8, 0.2 M NaCl), sonicated (3×20 s bursts at 16,000 nm λ) and insoluble material was removed by centrifugation (40,000*g*). Soluble protein was applied to a 5 ml Nickel Hi-trap column (GE Healthcare) in binding buffer (50 mM Tris-HCl pH 8 and 0.5 M NaCl) at a flow rate of 5 ml min$^{-1}$. Protein was eluted on a gradient of binding buffer to 50 mM Tris-HCl pH 8, 0.5 M NaCl and 1 M imidazole. Proteins were further purified by size exclusion chromatography using a Superdex 200 pg column (GE Healthcare) pre-equilibrated with 50 mM Tris-HCl pH 8 and 0.5 M NaCl. AhlA purification buffers also contained cOmplete protease inhibitors (Roche), and AhlA was stored in buffer containing 50 mM L-Arg and L-Glu[43], to prevent degradation. AhlB ran as a partially oligomerized species of 90 kDa on sodium dodecyl sulphate polyacrylamide gel electrophoresis (SDS-PAGE) when boiled with 1 mM DTT, but this high molecular weight species disappeared and AhlB ran at its expected molecular weight (36 kDa) when heated to 60 °C. This phenomenon may occur due to partial protection of the extensive hydrophobic domain of AhlB by SDS[44], but it was not observed for either of the AhlA or AhlC proteins.

Selenomethionine derivatized proteins were expressed in the same way, but prior to induction the culture was centrifuged for 10 min at 20 °C (30,000*g*) and the pellet was washed in 50 ml of minimal media (10.5 g/l K$_2$HPO$_4$, 1 g/l (NH$_4$)SO$_4$, 4.5 g/l KH$_2$PO$_4$, 0.5 g/l tri-sodium citrate, 5 g/l glycerol, and 0.5 g/l of each nucleobase adenine, guanosine, thymine and uracil, 1g/l MgSO$_4$·7H$_2$O, 4 mg/l Thiamine, 100 mg/ml of each of L-lysine, L-phenylalanine, L-threonine, and 50 mg/ml of each of L-isoleucine, L-leucine, and L-valine, and 40 mg/l of seleno-L-methionine). Cells were pelleted again, this time at 5000*g*, then resuspended in 500 ml of minimal media. The culture was then induced by addition of 1 mM IPTG, and grown for 2 days at 16 °C before harvesting. Selenomethione derivatized proteins were purified in the same way as nonderivatized proteins.

**AhlB and AhlC complex formation analysis.** Totally, 8 mg/ml of purified AhlB and 12 mg/ml of purified AhlC were mixed together in equal volumes and then applied to a Superdex 200 GE column which had been pre-equilibrated with 0.1 M NaCl and 50 mM HEPES pH 7.5. Fractions were collected and protein content was analysed by SDS-PAGE.

**Haemolytic assays.** The haemolytic activity of AhlA, AhlB and AhlC, was determined by measuring the release of haem from lysed cells photometrically at 542 nm, as described by Rowe and Welch[45]. A 0.25% (w/v) erythrocyte suspension was prepared from horse blood (Themo scientific) by washing cells repeatedly by centrifugation at 1500*g* for 5 min and resuspending cells in 10 mM phosphate buffer pH 7.4, 2.7 mM KCl and 137 mM NaCl. Varying concentrations of each AhlABC protein were incubated with 1 ml of the erythrocyte suspension and incubated on a blood wheel at 37 °C for 1 h. Erythrocytes were centrifuged at 1500*g* for 5 min, and the supernatant was removed for photometric analysis. A positive control of cells lysed in ddH$_2$O and a negative control with no protein added were used to normalise the data for 0 and 100% lysis. Haemolytic assays using AhlC head mutant were performed using the same method as used for the wild-type protein. For measurement of lysis against time, the rate of lysis was reduced in order to take measurements every 10 min, using a 0.5% (w/v) erythrocyte suspension with a protein concentration of 500 nM. All assays were carried out in triplicate.

**Ultracentrifugation.** Totally, 10 µM of purified AhlA, AhlB or 5 µM of purified AhlC were incubated with 45 µM N-heptyl-thioglucopyranoside at a total volume of 100 µl for 1 h at 25 °C. A third sample containing a mixture of 10 µM AhlB and 5 µM AhlC, or 10 µM AhlA, AhlB and 5 µM AhlC, was incubated with 45 µM N-hepty-thioglucopranoside at a total volume of 100 µl for 1 h at 25 °C. After incubation, samples were spun for 30 min at 214,500*g* in a Beckman Optima MAX ultracentrifuge at 4 °C. Supernatant was separated from the pellet and the pellet was resuspended in 100 µl 50 mM Tris-HCl pH 8 and 0.5 M NaCl. Finally, the soluble fraction and pelleted fraction from each sample was analysed by SDS-PAGE. In the ultracentrifugation assays AhlA partially degraded in the absence of other components, possibly due to disruption of the stabilising interactions between AhlA and the buffer components Glu and Arg, required to stabilise soluble AhlA.

**Liposome floatation assay.** Liposomes were prepared from *E. coli* total lipid extract (Avanti Polar lipids), using the extrusion method[46]. Totally 100 mg of solid

lipids were resuspended in 1 ml of 2:1 chloroform:methanol. A total of 100 µl of this solution was dried with nitrogen gas and then flash frozen in liquid nitrogen, before further desiccation under a vacuum for 2 h. The final lipid film was resuspended in 1 ml of 10 mM PBS pH 7.4 and vortexed until fully dissolved. The lipid solution was then extruded through a 100 µm filter 11 times to generate a uniform 100 µm liposome suspension, as gauged by negative stain electron microscopy. Totally, 300 µg of liposomes with 100 µg of protein for were incubated for 15 min at 37 °C before addition of 100 µg of protein 2 or the equivalent volume of PBS buffer. This mixture was then incubated for a further 45 min at 37 °C. After incubation the total volume was made up to 800 µl with 55% (w/v) sucrose in PBS. The liposome mixture was transferred to an ultraclear ultracentrifuge tube (Beckman Coulter), overlaid with 3.8 ml 40% (w/v) sucrose in PBS, followed by a layer of 400 µl PBS. Samples were centrifuged at 200,000×g for 4 h at 4 °C in a Beckman Coulter SW 55 Ti rotor. Six 100 µl fractions were taken from the top of the tube, 3.8 ml removed and a further six 100 µl fractions taken from the bottom, which were analysed by SDS-PAGE.

**Cross-linking of AhlC.** AhlC was cross-linked using both glutaraldehyde and 1-ethyl-3-(3-dimethylaminopropyl)carbodiimide hydrochloride (EDC) in a two step reaction. First 10 µl of 25% w/v glutaraldehyde was added to 90 µl of 1.4 mg/ml AhlC and incubated at room temperature for 15 min, to give 50% cross-linked AhlC, before buffer exchanging into 0.1 M MES pH 6 buffer using a Zebaspin 7K desalting column (sigma). Next 160 µg EDC was added to the partially cross-linked AhlC and incubated at room temperature for 30 min, to give fully cross-linked AhlC. This cross-linked AhlC was finally buffer exchanged into PBS buffer using a Zebaspin 7K desalting column for use in assays.

**AhlC crystallisation and structure determination.** Purified AhlC was concentrated to 40 mg/ml by centrifugation in a Vivaspin 10 kDa MWCO concentrator (Sartorius), and then buffer exchanged into 50 mM Tris-HCl pH 8, 100 mM NaCl with a Zebaspin desalting column (Sartorius). Native AhlC was crystallised at 16 °C by sitting drop vapour diffusion with 0.2 M MgCl2, 0.1 M Tris-HCl pH 7.0, and 10 % (w/v) PEG 10,000 (200:200 nl drop). Se-Met derivatized AhlC was crystallised using the same conditions, with crystals growing in a week. Prior to cooling in liquid nitrogen, crystals were cryoprotected in mother liquor containing an additional 20 % (v/v) ethylene glycol.

X-ray diffraction data were collected from a single crystal of Se-Met AhlC on beamline i04 of the Diamond Light Source a wavelength of 0.9790 Å. Images were integrated and scaled using FastDP[47]. The crystals diffracted to 2.8 Å resolution and belonged to the space group P6₅22, (Form 1, Table 1). Heavy atom sites and an initial map were calculated by the FastEP[48] pipeline. The asymmetric unit contained two chains of AhlC and the map was of sufficient quality to autobuild a model using Buccaneer (CCP4)[49,50], which optimised and completed in COOT[51], before refinement of the model against a higher resolution native dataset (2.35 Å), collected on the same beamline and integrated using Xia2 3dii[52]. (Table 1). Iterative rebuilding and refinement were carried out using COOT and REFMAC, respectively[51,53] to give a final model (PDB: 6H2E) with an R and Rfree of 0.22 and 0.27, respectively. Residues 231–241, and 268–274 of chain A and residues 155–160, 239–242, and 271–274 of chain B were omitted from the final model due to poor electron density.

The second crystal form of AhlC also crystallised in a week by sitting drop vapour diffusion in 0.2 M sodium chloride, 0.1 M tris pH 8, and 20% PEG6000 (200:200 nl drop). Crystal Form 2 was cryoprotected in mother liquor with an additional 20% ethylene glycol and crystals diffracted to 2.6 Å resolution and belonged to the space group P2₁. Data were collected on beamline i03 at the Diamond Light Source and processed by the Xia2 3dii pipeline[52,54], (Table 1). The structure was determined by molecular replacement using one subunit from the existing Se-Methionine AhlC structure as a search model. The asymmetric unit was comprised of four subunits of AhlC. Iterative rounds of manual model building and refinement were completed using COOT and REFMAC, respectively[51,53] to give a final model (PDB: 6H2D) with R and Rfree of 0.27 and 0.33, respectively. Residues 1–4 in all chains, and 71–89, 155–162, 229–248, 265–274 of chain P, 62–86, 147–170, 235–249, and 266–274 of chain Q, 68–89, 147–172, 230–274 of chain R, and 70–88, 158–164, 234–244, and 266–274 of chain S, were omitted from the final model due to poor electron density.

The triple L156T, L160T, L161T AhlC head mutant was crystallised using sitting drop vapour diffusion in 0.1 M imidazole pH8, and 10% Peg 8000 (100:100 nl drop). The crystal was cryoprotected in mother liquor with an additional 20 % ethylene glycol. The crystals diffracted to 1.92 Å resolution and belonged to the space group P 6₁22. Data were collected on beamline i04 at the Diamond Light Source and processed by the Xia2 Dials[55] pipeline (Table 1). The structure was determined by molecular replacement using one subunit from the existing Se-Methionine AhlC structure as a search model. The asymmetric unit was comprised of two subunits of AhlC With the tetramer assembled by crystal symetry. Iterative rounds of manual model building and refinement were completed using COOT and REFMAC[51,53] respectively to give a final model (PDB: 6R1J) with R and Rfree of 0.23 and 0.28, respectively. Residues 1–2 and 239–242 of chain J, and 1–3, 74–76 and 232–243 from chain D were omitted from the final model due to poor electron density.

**AhlB crystallisation and structure determination.** Purified AhlB was concentrated to 10 mg/ml by centrifugation using a Vivaspin 30 kDa MWCO concentrator (Sartorius), and then buffer exchanged into 0.2 M NaCl, 50 mM Tris pH8 using a Zebaspin desalting column (Sartorius). Crystals were grown by sitting drop vapour diffusion on microbridges in 60% methyl-2,4-pentanediol (MPD), 0.2 M ammonium phosphate and 0.1 M Tris pH8.5 (16 °C) (1:1 µl drop) by streak seeding with crushed microcrystals that had been diluted 10⁶-fold. Crystals grew in a month and were plunge-cooled in liquid nitrogen directly from the drop.

X-ray diffraction data were collected at a wavelength of 0.9790 Å on beamline i03 at the Diamond Light Source. Data were integrated to 2.94 Å resolution in space group C2 using the DIALS[55] pipeline and scaled and merged using Aimless[56]. Initial phases of AhlB Se-Methionine were obtained by SAD. Heavy atom sites and an initial electron density map were calculated using SHELXC,D,E[48], and an initial model was built using Phenix[57] followed by Buccaneer[50]. In this initial model, a ring of 10 AhlB subunits could clearly be seen. Rebuilding and refinement using COOT and REFMAC, respectively[51,53], resulted in a final model (PDB: 6GRJ) with R and RFree of 0.22 and 0.24, respectively. Residues 1–15, 337–360 and 202–208 of chains A, C, E, G and I, and residues 205–208 and 342–360 of chains B, D, F, H and J, were omitted from the final model due to poor electron density.

The second crystal form of the AhlB pore was grown by sitting drop vapour diffusion in a month, in 0.2 M ammonium phosphate, 0.1 M tris pH 8.5, and 50% MPD (200:200 nl drop). The crystals were plunge-cooled in liquid nitrogen directly from the drop and X-ray diffraction data were collected at beamline i02 at the Diamond Light Source and processed using Xia2 2da[52,54] in space group C222₁ to 2.55 Å (Table 1). The structure was determined by molecular replacement with PhaserMR[58] using the 10-mer Se-Methionine AhlB pore as a search model. Iterative rounds of refining and rebuilding with Coot[51] and REFMAC[53] were carried to give a final model (PDB: 6H2F) with R and Rfree of 0.19 and 0.28, respectively, with residues 202–208 in A and D, 200–208 in E and G, 205–207 in H and 342–367 in F, as well as residues 1–17 in B, D, F, H and J, and 339–367 in A, C, E, G and I omitted from the final model due to poor density.

The soluble form of AhlB was concentrated to 10 mg/ml using a Vivaspin 30 kDa MWCO concentrator (Sartorius), and buffer exchanged into 50 mM tris pH 8 and 0.2 M NaCl by Zebaspin column (Sartorius). Crystals were grown by sitting drop vapour diffusion in 0.2 M potassium thiocyanate, 0.1 M bis–tris propane pH 6.5, and 20% PEG3350 (200:200 nl drop). Crystals took a month to grow at 16 °C, and were cryoprotected in mother liquor containing an additional 20% ethylene glycol, and plunge-cooled in liquid nitrogen. Data were collected to 2.33 Å resolution at i03 of the Diamond Light Source, and processed in space group C2 using Xia2 3da[52], (Table 1). The structure was determined by molecular replacement in PhaserMR[58] using a single subunit of the AhlB pore with the hydrophobic head domain removed as search model to give a solution with three AhlB chains in the asymmetric unit. The molecular replacement solution model was further pruned to remove overlapping residues and an initial model was built using Buccaneer[50]. Refinement of the data and iterative rebuilding were carried out in Coot[51] and REFMAC[53], respectively, to give a final model (PDB: 6GRK) with R and Rfree of 0.22 and 0.25, respectively. Structural alignments described in this paper were undertaken using Dali[59].

**Electron microscopy.** Samples from lytic assays using erythrocytes, ultracentrifugation pellets, liposome flotation assays and prepared proteoliposomes were all visualised using negative stain TEM. Proteoliposomes were generated by incubating 10 µg of Ahl proteins with 20 µg of fresh liposomes in 100 µl of 10 mM PBS buffer pH 7.4, at 37 °C for 1 h and kept on ice before use. Totally, 5 µl of each of the respective samples was pipetted onto a glow discharged (copper 300 mesh) carbon-coated grid, and then stained with 1% (w/v) uranyl formate. Carbon grids were then air dried before storing for up to 5 weeks at room temperature. Electron micrographs were collected using a Philips CM100 100 kV transmission electron microscope, equipped with a Gatan 1 K CCD camera. Micrographs were collected with a pixel to nm ratio of 0.72 pixels per nm, and this was used in the calculation of the pore sizes.

**Modelling of AhlA.** A homology model of AhlA was generated using Phyre2[60] and the structures of soluble MakA (pdb: 6EZV), Hbl-B (pdb: 2NRJ) and XaxA (pdb: 6GY8) as templates. The model of AhlA contained 312/354 residues (84%) at a Phyre2 accuracy of >90%.

**Generation of figures.** All protein cartoon diagrams and surface rendering used in figures and movies were made using PyMOL[61], with intermediates in the movie morph generated using LSQmann[62].

**Reporting summary.** Further information on research design is available in the Nature Research Reporting Summary linked to this article.

## Data availability

Data supporting the finding of this manuscript are available from the corresponding author upon reasonable request. Atomic coordinates for AhlC Form 1 (PDB code 6H2E), AhlC Form 2 (PDB code 6H2D), AhlC Head mutant (PDB code 6R1J), AhlB soluble

(PDB code 6GRK), AhlB pore SeMet (PDB code 6GRJ) and AhlB pore Form 2 (PDB code 6H2F) have been deposited in the RCSB Protein Data Bank. The source data underlying Fig. 1a–d, f, g and Supplementary Figs. 3a, b and 11 are provided as a Source Data file.

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

## Acknowledgements

We would like to thank Julia Pridgeon of USDA for the kind gift of *A. hydrophila* genomic DNA, the Diamond Light Source for access to beamlines I02, I03 and I04 that contributed to these results, and the beamline scientists for their assistance with data collection. Electron Microscopy and mass spectrometry were performed in the Faculty of Science Electron Microscopy Facility and the ChemMS Mass Spectrometry Centre, at the University of Sheffield. This work was supported by BBSRC Doctoral Training Grant BB/F016832/1 awarded to J.S.W., University of Sheffield scholarships awarded to J.S.W. and A.M.C.-A., BBSRC grant BB/D524975/1 and EPSRC grant EP/L026872/1. We dedicate this paper to Peter Artymiuk (decd.) who initiated this work in Sheffield.

## Author contributions

P.J.B. conceived the project; P.J.B., A.M.C.-A., J.S.W. and C.B. designed the experiments; A.M.C.-A., J.S.W., S.E.S., S.B.T. and S.P.D. performed the experiments; P.J.B., A.M.C.-A., J.S.W., C.B., J.B.R. and P.A.B. interpreted the data and P.J.B., A.M.C.-A. and J.S.W. prepared the paper with input from all authors.

## Additional information

**Competing interests:** The authors declare no competing interests.

