## [Peer Review File · Nature Communications]

Reviewers' comments:

Reviewer #1 (Remarks to the Author):

The manuscript by Wilson et al. characterizes the tripartite α -PFTs from *Aeromonas hydrophila*, which they found by bioinformatical search using the sequence and structure of *B. cereus* NheABC. After cloning and purification they characterized the functional activities by haemolytic assays with horse erythrocytes for AhIA, AhIB, AhIC, binary combinations of them, and all together and show that the tripartite system has the highest activity (95%). On the other hand, the activity of the binary system AhIB/AhIC has only ~50% activity. Using ultracentrifugation experiments they show the interaction of AhIA/B/C proteins solubilized in detergent. Further they solved the crystal structures of AhIB and AhIC. The structure of AhIB was solved in one soluble and two pre-pore forms. AhIC crystallized in two forms as soluble tetramers. Both, the structures of AhIB and AhIC, were solved ab initio by Semet-SAD. Like for many other α -PFTs, they observe a large conformational change between the soluble AhIB monomers and the oligomeric "pre-pores" where AhIB was found to adapt two different subunit conformations (Type 1 and Type 2) within the homo-oligomeric 10mer ring.

The authors present here the structures of an interesting pore forming toxin components, which they identified by extensive bioinformatic search. The experiments (bioinformatics, purification and crystallization) were performed well and structures seem to be convincing. However, the "story" the authors present here is far from being complete. To be published in Nature communications the authors should address the following points:

How the authors can verify, whether the oligomeric 10mer ring seen in the AhIB crystal structure represents either the pre-pore or the pore form?

Are the oligomeric structures seen in Figure 4 formed in the absence of liposomes or erythrocytes as well? Are the liposomes (or erythrocytes) essential for pore or pre-pore formation?

Did the authors looked for AhIABC pores in absence of liposomes?

It is shown that AhIB/AhIC and AhIA/AhIB/AhIC form pores on the horse erythrocytes used for the activity measurements. Why the authors don't show these pores in negative-stain EM, first to compare them to the pores seen on the liposomes and to compare the AhIB/C pores to the AhIA/B/C pores (or pre-pores). As AhIB forms the homo-oligomeric 10mer rings, they should be also included in the comparison to verify that the crystal structure of the "pre-pore" ring really corresponds to the "real" pores seen on the erythrocytes.

What is the experimental evidence for the absence of AhIB type1 conformation in the "pore" complex of AhIBC? (Figure 8c).

Is it possible to form/isolate the AhIBC complex in vitro to proof that AhIC is "added" to the (now) homo-oligomeric AhIB pore?

Similarly, what is the basis for the AhIABC pore shown in Figure 8c? According to this model AhIA would constrict the pore, but it is mentioned that the N-terminus of AhIB folds towards the pore center (Supplementary figure 6).

If the 15(?) residues of the AhIB N-terminus are able to block the AhIB pore, how the authors explain the possibility that AhIA could fit into the pore center as shown in the last model in Figure 8c?

How the authors could exclude or verify the idea that AhIC represents one of the subunit conformations found for AhIB in the crystal structure?

Did they try to model AhIC to AhIB type1 or 2 subunit conformation?

Is it possible for the authors to show the presence of AhIA in the oligomeric ring structures of AhIB alone or AhIB/C?

Minor comments:

Lines 201-203: It is shown that MPD induces the oligomerization of α -HL, not its pore formation. The cited literature describes the x-ray structure, whereas the correct citation should be Tanaka et al. 2011.

The structures of XaxAB recently published by Schubert et al. (<https://doi.org/10.7554/eLife.38017>) should be included in the introduction and discussed in the text.

Figure 1e: Could you give an explanation why the signal for the pellet of AhIB+AhIC is much higher than the signal for the pellet of AhIA+AhIB+AhIC? Has AhIA an positive effect on the solubility of the AhIB+AhIC complex?

Figure 2c: For better overview, the shown molecules should be in the same orientation like the side views in Figure 2a and 2b.

Figure 8: The region of AhIX proteins inserted in the figures do not correspond the hydrophobic region shown in Figure 2b.

Supplementary Table 1 is missing.

Supplementary Figure 2: In 2a, AhIB is the main component in the SDS-PAGE analysis. In 2c, however, the purity of AhIB in lane "B" is suboptimal and by far worse than in 2a. Are the lower bands for AhIB in 2c an indication for protein degradation?

Supplementary Figure 2c: How the authors explain the low intensity for the AhIC signal? They mix AhIB and AhIC in 1:2 ratio and one would expect ~80-100 mAu for the signal of AhIC, but it is almost in the noise level.

Supplementary Figure 3 is redundant (same as Figure 1e) and therefore unnecessary.

Reviewer #2 (Remarks to the Author):

The manuscript by Wilson and colleagues describes structural and functional features of tripartite pores formed by a proteins AhIA, B and C from bacteria *Aeromonas hydrophila*. The authors have used bioinformatics approach in order to search for novel representative of cytolysin A (ClyA) family of pore forming toxins by using a known member as a probe. They have discovered a representative in *A. hydrophila* and have found additional components in its genome. They have expressed all three components in *E. coli* and performed structural and functional analysis with recombinant proteins. Based on the structural and functional data they propose a model of pore assembly, which is a novel for the ClyA family.

The novelty of the presented work lies in the fact that the pore is formed by the three components and not one (as in the case of ClyA) or two (as in the case of YaxAB). The structural features of AhIA and B components that are studied here show great resemblance to other known members of the ClyA family, with some exceptions, so this is not particularly novel aspect of the work. The authors also show some TEM data, but they are not of the sufficient quality to draw any firm conclusion about the pore structure. In fact, the positioning of AhIA component in the pore and the model of AhI pore assembly is not adequately supported by the presented data and requires further work.

Comments:

Line 147: Supplementary table 1 is just the list of proteins identified after the Blastp search by using AhIB as a probe. The results for AhIA and C are mentioned (lines 143-144), but actual sequences are not presented. I propose to put hits for all three components in Supp. table 1 and organise this table appropriately.

The SEC data on Supplementary figure 2 shows additional peak at app. 42 ml for B and C and this peak, quite significant, is left unaccounted for by. Also, it is not clear which fractions are shown on the SDS-PAGE gels, is this peak included? What does the CFE and Ni-HP mean on the gels? The molecular weight markers are used but not specified to which molecular weights bands correspond to. The chromatograms and gels should be better annotated.

It seems that the B component is different in a and c panels. While it is quite pure in peak2

fractions in a, it has a lot of impurities present in panel c gel. Please explain.

This figure also shows the result for 1:2 mixture of B and C components, but the ratio between the peaks does not correspond to 1:2 ratio (presuming they absorb to a similar extent). This should also be clarified.

Panels c and d are mixed in Figure 1.

Line 186: here also Supplementary Figure 3 should be cited. Now it is left unaccounted for by. In general, some supplementary figures are not cited in order they appear.

Line 190- what is the reason for AhIA degradation in presence of detergents?

It is not clear why there is a need to use detergents in experiment presented in Figure 1e (line 195). The detergent micelles are poor substrate for membrane environment, presuming the reason was to test the ability of proteins to bind to lipid membranes. Why were not appropriate lipid vesicles used, which could better report the ability of individual components to associate with lipid membranes in isolation or when in complex. I think this panel has low relevance to the in vivo situation.

To better understand explanations of the structure of pre-pore AhIB in the text I propose to include another panel on Figure 2, where different parts of the pre-pore AhIB (i.e. helices, head, tail etc.) would be highlighted, similar to the one shown for the soluble AhIB on Figure 5a.

Line 201: on what basis the authors conclude that the solved structure of AhIB is a membrane bound form. Without showing some evidence this should be rephrased.

Lines 213 and 214: alpha helices 3 and 4 could also be labeled in the left image of panel a (Figure 2). Also the head and tail regions of the protein, which are referred to in lines 2019 and 211, could be highlighted in the panels of Figure 2a.

The features of soluble AhIB, in particular the beta-tongue with hydrophobic residues should be compared with other members of ClyA family. Is there a common mechanism present in the whole family? The comparison could include structural models from different representatives with the hydrophobic part labeled, possibly as a supplementary figure.

Line 321: it is claimed that no membrane-bound assemblies of AhIC alone could be observed in electron micrographs. Why are these data not presented in Figure 4?

The observation that some of liposomes are saturated with pores and others remain free of pores (lines 328-329) requires an explanation. Could the reason be high cooperativity of the membrane assembly process of AhI pores?

The quality of the presented data (Figure 4) does not clearly show that tetramer of AhIA cannot bind the membrane. I.e. the proposal that AhIA tetramer must dissociate in order to bind the membrane (line 378) should be supported by some functional data.

Lines 392-399 explain the results of the functional mutants assay. This should better fit to the results section.

Why were E.coli lipids used for proteoliposome preparation? Is this a natural substrate for an AhI system? Is there any lipid specificity associated with the binding to the lipid membrane by any of the AhI components?

Where is evidence that pores in Figure 4e are formed by all three components a, b and c? Without further analysis of images and additional structural details of the pore it is not possible to draw any firm conclusions.

Is there any functional data consistent with the model proposed in Figure 8? The authors could try

using size markers such as fluorescent dextrans in order to get an idea about the pore diameter of various AhI assemblies, or perform ionic current measurements in a planar lipid membranes setup.

Reviewer #3 (Remarks to the Author):

The manuscript describes a novel tripartite alpha pore-forming toxin (PFT) system, AhIABC, with functional and structural analogy to the already described ClyA type toxin. The authors present structures of two out of three components of the toxin system, with the membrane-permeating component in two conformations. Based on structural and functional data and also a site-directed mutagenesis study, a model of holotoxin pore assembly is proposed.

The presented data support a pore formation reaction analogous to the described unimolecular or bimolecular alpha-PFTs. The authors demonstrate well that only a holotoxin made up of all three components form a fully functional pore, underlining the novelty of their finding. Altogether, the results and the novelty of the three-component alpha-PFT make the paper interesting for a publication in Nature communication.

The fact that five different structures (where 2x 2 show the identical state, if I'm not mistaken) combined with the fact that two conformations are present in the AhIB pre-pore make the manuscript confusing. I recommend to present the individual structures in a more accessible way by clearly writing what the two different structures of the AhIB pre-pore show, what they have in common and what are the differences (if any), analogous to the two AhIC structures. The storyline often goes forward and back in the figures, which makes it a bit difficult to read. That could be improved.

Furthermore, the pore assembly model should be strengthened by more functional and mutational studies, please see below.

Altogether, when the manuscript would be complemented with the suggested experiments to strengthen the assembly hypothesis and the storyline would be presented in a clearer way, I recommend publication.

Suggested additional experiments to strengthen the content of the paper

(1) Hemolysis kinetics assays (see. l.167ff and Fig.1) to get more information about the cell lysis behavior of individual components, especially after pre-incubation of non-lytic components with erythrocytes and addition of other components of the holotoxin. Is the activation then fast or slow, with or without a pronounced lag time? That would give information about the timing of assembly and conformational transitions. The experiments could also help to support or disprove the assembly model with AhIC as initial membrane-bound subunit.

(2) The UC experiments show only that large assemblies that precipitate in an UC run are formed. Evaluation of that complexes via gel filtration and electron micrographs would give more information. Are similar complexes formed like observed in Fig. 4? (see l.182ff).

(3) Site-directed mutagenesis: Mutagenesis experiments on the AhIC head domain showed the loss of hemolytic activity. Mutagenesis on the hinge regions of AhIB would strengthen the assembly and conformational transition model, analogous to experiments described for YaxB and ClyA.

Point-by-point analysis of the manuscript:

Introduction:

l. 57: "initially fold into": better write "are produced as" to avoid confusions

l. 76: A recent publication also compares the XaxAB pore with the respective monomer structures. The XaxAB pore shows more details than the YaxAB pore and stresses the fact that pores with different assembly numbers can be created.

l. 109: "the AhIC tetramer dissociates": It is unclear to me what is meant here, where or from what it dissociates. Please write to be more clear.

I. 110: "large scale conformational change": How large? A brief comparison with the changes of ClyA and YaxB would be helpful here.

I.111: modeling of AhIA: That suggests a homology model that is presented together with the crystal structure. However, this is not the case here.

Results:

General arrangement of description of structures and figures:

I find it a bit confusing to start with the AhIB pre-pore, then show them on the membrane and then describe the monomer structure. References to figures are then going from 2 to 3 to 4 and then back to 3. Maybe re-arrange the figures 3-5 to follow the storyline better.

I.144ff and SI Table 1: In which homology group was the single component ClyA found? Please state in the text, if not found at all, please indicate why it was not identified as a homolog.

I.166f: better wording in my opinion would be: "..., indicating that indeed all three components of the AhIA/AhIB/AhIC toxin system are required for its maximum activity."

I.167 and Fig1: An experiment that does not only show the end product after a distinct incubation time, but an on-line measurement of lysis kinetics (like described for ClyA) would give us more information regarding the lysis behavior with the different components. It would also enable the possibility to pre-incubate erythrocytes with different components and add the 3rd component later. If e.g. there is no lag time anymore when adding AhIA to pre-incubating AhIBC, it would show that no large conformational change is necessary for that component.

I.174ff: Fig 1d for AhIC, Fig 1c for AhIB?

I.171ff: What happens if AhIA, B or C are used in excess over the other components? Can you show also data points available for them? Do the curves for AhIA and AhIB reach saturation at a 1:1:1 ratio or is there a further increase (or even decrease, like NheABC)?

I.180: Please describe briefly what happens in NheABC and speculate why AhIABC is different.

I.182ff, UC experiment: In that experiment it is only shown that the proteins are in the pellet at high-speed centrifugation, nothing more. As you have the possibility to visualize the samples in negative staining, do you see large symmetric assemblies that are consistent with a possible pre-pore or the pore? Gel filtration on Superose 6 could also give more information regarding the size of the complex, if it is aggregate or a species with a defined size. IN Fig. 1e it is also obvious that only ~20% of the trimeric complex are in the pellet, while about 50% of the BC complex are in the pellet. Any possible reasons why?

I recommend additional experiments (neg stain EM, gel filtration) to specify the type of large complexes.

I.210ff, indicate neck, tail etc. in Fig.2; state what is outer and inner diameter of the pore

I.218: indicate Phe and Tyr in the same Fig panel (Y245 is indicated in Fig 2A, F203 in Fig 2B)

I.221ff: Two conformations of AhIB in the oligomeric structure and two different AhIB (pre-)pore structures solved – that is confusing. What is the difference between both crystal structures? Why is one structure named pre-pore and the other one pore? Please make that clear.

I.247ff: The AhIB structure's head looks strikingly similar to the ClyA monomer and also the HbIB structure. Please refer to that and maybe compare them briefly. It might be useful for a better understanding of the toxin system to stress here that AhIB is that part of the trimeric toxin that is most similar to ClyA.

I.280ff.: Can you provide information that the arrangement of the tetramer in the crystal package is consistent with the real situation (e.g. cross-linking studies or negative stain images)? How would the tetramer dissociate in pore formation? Are there examples in the literature where a Leu-Zipper is dissociated by membrane contact interactions? Does the tetramer dissociate in a detergent solution?

AhIC seems to be the toxin subunit that is consistent with The A subunit of binary toxins, please indicate that here. Also an additional figure panel comparing YaxA and AhIC (as well as YaxB and AhIB) would be helpful.

I.290: How significant is the subunit difference in AhIC in comparison to AhIB?

I.299: K151 is not shown in the figure.

I.329: To underline the finding that there are liposomes with and without pores, an additional Supplementary Figure with several images of liposomes would be helpful.

Discussion:

I.356ff: A supplemental figure comparing the conformational transitions of ClyA, YaxB and AhIB would be helpful.

I.392ff: The chapter describes experimental results, I think it would be better in the "Results" section.

I.434ff, model of toxin assembly: If the model is correct, it should be possible to identify AhIC bound to the membrane alone, either via EM by analyzing liposome and soluble fractions in solution via SDS-PAGE (analogous to Fig.1e). Are EM images or biochemical experiments showing the co-localization of AhIC with membranes available to show?

I.463ff: It should also be stated here that the AhIABC pores appear larger than the AhIBC pores. The side view in Fig.4e is definitely larger than the top views in Fig. 4d suggests. If possible, a low-resolution 3D reconstruction of more images could be compared with the YaxAB or XaxAB pore to see size differences and also show the presence of an additional molecule.

L468ff. A figure showing NheA and the proposed domains interacting with the AhIB analogon and the TM regions would be helpful. Does the sequence identity allow the building of a homology model of AhIA?

I.489ff: The last chapter of the discussion gives a very good overview of the differences and common features of the mono- bi- and tripartite toxin complexes. It could be complemented by a figure showing schemes of the individual components as monomers and assembled.

Methods:

I.563ff: Something seems to be missing in that sentence, does it mean AhIA was stored in buffer with 50mM Arg and Glu?

I.572ff: Were the non-Se-Met expressions also performed in minimal medium? It is not stated in the section above.

I584: Co-purification as heading is misleading, better: "Complex formation analysis"

I.602: Is there any reason why a 1% erythrocyte suspension was used for the AhIC head mutant instead of 0.25%? Are the results comparable with respect to stoichiometries of erythrocytes and toxin?

Crystallization: To be consistent, maybe also describe crystallization and structure determination of the AhIB monomer before the oligomer, like it was done for AhIC. For AhIB monomer, crystal growth take a month – how long did it take for the others?

What re the differences between the two AhIB pore structures?

Figures:

Fig1 b-d: Panels c and d are referred wrongly in the text.

Fig. 2: The same colors for the same type of protomers and the lack of depth of focus make it difficult to identify and trace different protomers, especially in the left and middle view. Please improve the figure, e.g by using color gradients for the protomers (e.g. light green to dark green and light orange to brown for the two types of protomers). Also shadows, border lines to the models and depth of focus like in the right panel would be helpful.

In panel b, please indicate which views are shown (section/side view).

A panel comparing the two types of protomers side-by-side is missing and would be helpful. It would be also helpful to indicate the different regions and crucial residues in that panel (text I. 210ff).

Fig.3: color the two hinges with different colors, show the Figure after the individual Figure of the AhIB monomer (Fig.5).

Fig.4: It would make it easier for the reader to indicate the contents of the liposomes also in the figure itself, not only in the legend. Indicate pore top views also in panels d and e. It might also be better for understanding the storyline of the text to split that figure in parts that complement the respective figures of the individual components.

Fig.6: re-arrange, like the storyline is described in the text (first description of AhIC, then comparison with AhIB). A scheme of the Q subunits is missing in panel B.

Fig.7: b,c: Is one AhIB replaced by an AhIC? Please indicate that in the text and Fig legend, and also label the subunits in the Fig. panel.

Fig.8: AhIA missing in the scheme in b

What's the physiological relevance of the AhIB "pre-pore"? For me it appears as an artificial off-pathway construct that cannot be recovered to form a fully active AhIABC pore. Therefore, the term "pre-pore" is misleading.

A side view of the proposed AhIABC pore in the membrane would be helpful to understand the positioning of AhIA.

Supplementary 2: Indicate the sizes of the proteins in panel C. It looks like the mass difference between AhIB and AhIC is much more than only 8 kDa because of the many marker proteins between them. Contrast of gels in a and b is very low. Indicate marker protein sizes in all gels. Supplementary 5: AhI instead of Ah. Use colors with more contrast for map and model, also in Suppl. Fig 4a,b. IN both Suppl. Fig 4 and 5, please indicate from which part of the structure the maps are shown.

Suppl. 6: An additional side view showing where the disordered density is located would be helpful.

Suppl. Fig. 10ff: 2x Figure 11? The overlays are sometimes difficult to interpret. In my opinion it would be more helpful to have the two structure schemes not overlaid, but side-by-side.

Tables:

Table 1 resolution: Some values are given as low-high, others high-low. Please change them to be consistent.

Point by point response to referees comments – manuscript NCOMMS-18-23936.

Identification and structural analysis of the tripartite α -pore forming toxin of *Aeromonas hydrophila*.

We would like to take this opportunity to thank the reviewers for their careful reading of our paper and their comprehensive comments. The reviewers have suggested that a number of extra experiments are required to add to the results being presented. We agree with their suggestions and have completed the experiments and incorporated these major revisions into the manuscript as detailed below:

Reviewer #1 (Remarks to the Author):

*The manuscript by Wilson et al. characterizes the tripartite α -PFTs from *Aeromonas hydrophila*, which they found by bioinformatical search using the sequence and structure of *B. cereus* NheABC. After cloning and purification they characterized the functional activities by haemolytic assays with horse erythrocytes for AhIA, AhIB, AhIC, binary combinations of them, and all together and show that the tripartite system has the highest activity (95%). On the other hand, the activity of the binary system AhIB/AhIC has only ~50% activity. Using ultracentrifugation experiments they show the interaction of AhIA/B/C proteins solubilized in detergent. Further they solved the crystal structures of AhIB and AhIC. The structure of AhIB was solved in one soluble and two pre-pore forms. AhIC crystallized in two forms as soluble tetramers. Both, the structures of AhIB and AhIC, were solved ab initio by Semet-SAD. Like for many other α -PFTs, they observe a large conformational change between the soluble AhIB monomers and the oligomeric “pre-pores” where AhIB was found to adapt two different subunit conformations (Type 1 and Type 2) within the homo-oligomeric 10mer ring.*

*The authors present here the structures of an interesting pore forming toxin components, which they identified by extensive bioinformatic search. The experiments (bioinformatics, purification and crystallization) were performed well and structures seem to be convincing. However, the “story” the authors present here is far from being complete. **To be published in Nature communications the authors should address the following points:***

- 1. How the authors can verify, whether the oligomeric 10mer ring seen in the AhIB crystal structure represents either the pre-pore or the pore form?*
We agree with the referee that referring to the AhIB 10-mer oligomer as a pre-pore, infers a direction of fully active pore assembly, and is thus somewhat confusing. New time course assays (Figure 1; lines 194-216) show that if erythrocytes are pre-incubated with AhIC prior to addition of AhIB, then lysis is immediate, whereas when erythrocytes are pre-incubated with AhB prior to addition of AhIC, only limited lysis is achieved. Thus either there is a very slow addition of AhIC to the AhIB pores, to form a lytic pore, or that AhIC cannot be added at all to AhIB oligomers, and that fresh AhIBC pores are formed from the remaining soluble AhIB component, in agreement with *B.cereus* Nhe and Hbl toxins where the equivalent components to AhIC (NheC or Hbl-B) have been shown to be the first priming step (lines 436-440). We have thus referred to the AhIB 10-mer oligomeric pore structure in the revised manuscript as a pore-form rather than a pre-pore form.

2. *Are the oligomeric structures seen in Figure 4 formed in the absence of liposomes or erythrocytes as well? Are the liposomes (or erythrocytes) essential for pore or pre-pore formation?*
No pore-like oligomers of AhIB, AhIBC or AhIABC can be identified by EM or gel-filtration in the absence of erythrocytes, liposomes or detergent. (Supplementary Figure 2c & 3c, lines 169-175; 230-232).
3. *Did the authors look for AhIABC pores in absence of liposomes?*
Yes and none can be found. We have undertaken new liposome float assays (Lines 230-232 and Supplementary Figure 3c), analyzing both liposome bound and liposome free fractions, which show no indication of pore like structures in the liposome free fractions.
4. *It is shown that AhIB/AhIC and AhIA/AhIB/AhIC form pores on the horse erythrocytes used for the activity measurements. Why the authors don't show these pores in negative-stain EM, first to compare them to the pores seen on the liposomes and to compare the AhIB/C pores to the AhIA/B/C pores (or pre-pores). As AhIB forms the homo-oligomeric 10mer rings, they should be also included in the comparison to verify that the crystal structure of the "pre-pore" ring really corresponds to the "real" pores seen on the erythrocytes.*
We have repeated the negative-stain EM experiments using erythrocytes instead of liposomes (Figure 1, Lines 211-217). In all cases the same structures are seen using both types of lipid bilayer, indicating that the pore like structures observed are those present *in vivo*. The AhIB pores observed in liposomes and erythrocytes are of the same dimensions as the crystal structure, indicating that the crystal structure corresponds to that of the lipid-bilayer bound structure (Lines 274-280, Figure 3a).
5. *What is the experimental evidence for the absence of AhIB type1 conformation in the "pore" complex of AhIBC? (Figure 8c).*
We have compared the structures of both Type 1 and Type 2 pore-conformations of AhIB to the single conformation of the structures of the B component of the bipartite toxins YaxAB and XaxAB. In both cases AhIB Type 2 is closer to that seen in YaxB and XaxB (rmsd C α 3.5Å, 3.5Å), than AhIB type 1 (rmsd 4.3Å, 3.8Å) (Supplementary Figure 7; lines 304-321). In addition, the arrangement of the tail helices of Type 1 are in a similar arrangement to those seen in the soluble structures of ClyA, NheA, HblB and MakA, whereas in Type 2 of AhIB, the tail helices adopt the same conformation as seen in the pore form of YaxB and XaxB. We have modeled both AhIB and AhIC onto the structure of the decameric YaxAB pore. If this model is constructed from a mixture of Type1 and Type 2 AhIB conformations, then there is severe overlap between the subunits, but a model constructed from Type 2 AhIB and AhIC can be built in the same organization as YaxAB without clashes, indicating that the AhIBC pore structure is most likely constructed from the AhIB type 2 conformer (Lines 485-500, Supplementary Figure 14). Structural comparisons between AhIC and the two conformers of AhIB show that AhIC is most similar to the Type 2 conformations (rmsd 2.4Å, Figure 6, Lines 345-348; 458-463), and would thus pack more closely in an AhIBC oligomeric pore. In addition, the Type 2 conformation can be arrived at by more rotation around the same hinges that describe the change from soluble to Type 1 forms, and thus Type 1 appears to be an intermediate on the conformational change from soluble AhIB to pore form AhIB. (Supplementary Movie 1 and Figure 4, lines 304-321). Taken together, this suggests that the Type 2 conformation of AhIB is most likely

that seen in the active pore.

6. *Is it possible to form/isolate the AhIBC complex in vitro to proof that AhIC is “added” to the (now) homo-oligomeric AhIB pore?*

The new time course assays that we have completed (Figure 1, lines 194-217) show that lytic activity is much higher if AhIB and AhIC are mixed together with erythrocytes or if AhIC is pre-incubated, before addition of AhIB, compared to when AhIB is pre-incubated with erythrocytes before addition of AhIC. This implies that it is more favourable for the AhIBC pore to form from both components simultaneously, rather than C being added to a pre-formed AhIB pore. The text has been altered to make this clear.

7. *Similarly, what is the basis for the AhIABC pore shown in Figure 8c? According to this model AhIA would constrict the pore, but it is mentioned that the N-terminus of AhIB folds towards the pore center (Supplementary Figure 6).*

If the 15(?) residues of the AhIB N-terminus are able to block the AhIB pore, how the authors explain the possibility that AhIA could fit into the pore center as shown in the last model in Figure 8c?

We agree that the proposed model of the AhIABC pore in Figure 8 is not clear. Negative stain images of AhIBC and AhIABC pores (Figure 1, lines 211-217) show they are of a similar size and much bigger than the AhIB pores alone. We have modeled a proposed AhIABC pore using the structures of the XaxAB and YaxAB pores as a template. YaxAB and XaxAB adopt a funnel shaped structure, with sufficient space in the tail regions for AhIA to be included between the AhIB and AhIC subunits in the complete pore, without a significant increase in size of the tail. There would be an increase in the diameter of the head, to retain the same internal channel diameter through the membrane, but as the diameter of the pore is much smaller at the head compared to the tail, then this increase would not be visible on the negative stain EM images. In addition, the crystal structures of the AhIB pores show that the head helices can adopt different positions within the 10-mer oligomeric pore, and thus it is likely they could move to accommodate AhIA in the AhIABC lytic pore. (Supplementary Figure 6, Lines 298-302). The role of the disordered N-terminal residues of AhIB is currently unclear, but we note that the equivalent residues are also disordered in the YaxB and XaxB components of the active pore structures of YaxAB and XaxAB. (Lines 333-334). We have altered Figure 7 that describes schematics of the pore formation, to make this more clear.

8. *How the authors could exclude or verify the idea that AhIC represents one of the subunit conformations found for AhIB in the crystal structure?*

Did they try to model AhIC to AhIB type1 or 2 subunit conformation?

The rmsd between AhIC and pore-form AhIB conformers show they have similar structures (Figure 6). Superposition of a AhIC subunit onto a single Type 2 AhIB subunit in the 10-mer pore, shows that the residues on the interface between the AhIB type 1 and type 2 interface are conserved in AhIC and thus similar interactions could occur. We have altered Figure 6 and the text (Lines 345-348; 459-464) to make this clearer.

9. *Is it possible for the authors to show the presence of AhIA in the oligomeric ring structures of AhIB alone or AhIB/C?*

We have undertaken new liposome float assays, which show that AhIA, AhIB and AhIC all bind to liposomes when added together. Negative stain EM images of the floated liposomes show the presence of pore structures (Figure

1, lines 219-235), which have a larger diameter (16 nm) than the AhIB pores alone (10 nm). In addition, ultracentrifugation of AhIA/AhIB and AhIC with detergent shows that assemblies > 400kDa containing all three components are formed. Taken in combination with the assays, which show that AhIA is required for maximal lysis, the modeling studies described above and the fact that the AhIB or AhIBC pores are hydrophobic, whilst all other ClyA family pores are lined with hydrophilic residues, there is compelling evidence that AhIA must be a component of the fully lytic pore.

Minor comments:

10. *Lines 201-203: It is shown that MPD induces the oligomerization of a-HL, not its pore formation. The cited literature describes the x-ray structure, whereas the correct citation should be Tanaka et al. 2011.*

Agreed, text has been changed and reference added (Line 287).

11. *The structures of XaxAB recently published by Schubert et al. (<https://doi.org/10.7554/eLife.38017>) should be included in the introduction and discussed in the text.*

The XaxAB structures had not been published when we submitted our manuscript. We agree that they are important in describing the ClyA family and should be included in the revised manuscript. We have changed the introduction (Lines 78-89; 116-124) and also referred to these structures throughout the manuscript as appropriate (Figures 4, 7; Supplementary Figures 7, 13)

12. *Figure 1e: Could you give an explanation why the signal for the pellet of AhIB+AhIC is much higher than the signal for the pellet of AhIA+AhIB+AhIC? Has AhIA an positive effect on the solubility of the AhIB+AhIC complex?*

In the ultracentrifugation experiments using all three components, all of the AhIA added is pelleted with a similar amount of AhIB and AhIC, perhaps indicating that only ABC pores have been formed and that no AhIBC pores have formed. As neither ABC or BC complexes form in the absence of bilayer or detergent, this implies a preference for the formation of ABC pores over BC pores. The referee is correct in suggesting that the formation of the AhIABC complex in detergent appears to result in a very much reduced amount of the AhIBC complex and this has now been discussed in the results/discussion section (Lines 219-235; 509-517 Figure 1)

13. *Figure 2c: For better overview, the shown molecules should be in the same orientation like the side views in Figure 2a and 2b.*

We agreed and have changed the figure so that all molecules are shown in the same orientation, which has also been renumbered to Figure 3.

14. *Figure 8: The region of AhIX proteins inserted in the Figures do not correspond the hydrophobic region shown in Figure 2b.*

We have changed the figure (now Figure 7) so that the hydrophobic regions correspond to those in Figure 3 and the correct hydrophobic regions are inserted in to the membrane.

15. *Supplementary Table 1 is missing.*

Supplementary table 1 was included, but is very large and this table now also contains BlastP results for AhIA and AhIC as requested by reviewer 2.

16. *Supplementary Figure 2: In 2a, AhIB is the main component in the SDS-*

PAGE analysis. In 2c, however, the purity of AhIB in lane “B” is suboptimal and by far worse than in 2a. Are the lower bands for AhIB in 2c an indication for protein degradation?

In the figure (Supplementary Figure 2) all samples have been boiled before loading on to the SDS-Page as described in the methods section (Lines 630-635). AhIB partially forms a large aggregate when boiled. The lower band is the monomeric AhIB, no stack was used on this gel explaining the smearing. The Figure has been relabeled to make this clear and the legend has been altered to reflect this (Supplementary Figure 2).

17. *Supplementary Figure 2c: How the authors explain the low intensity for the AhIC signal? They mix AhIB and AhIC in 1:2 ratio and one would expect ~80-100 mAu for the signal of AhIC, but it is almost in the noise level.*

The relative absorbance of AhIC to AhIB is very low because AhIC does not contain any Trp residues and thus the extinction coefficient is very small ($1490 \text{ M}^{-1}\text{cm}^{-1}$) compared to that of AhIB ($25440 \text{ M}^{-1}\text{cm}^{-1}$). We have changed the Figure legend to reflect this (Supplementary Figure 2).

18. *Supplementary Figure 3 is redundant (same as Figure 1e) and therefore unnecessary.*

We agree and have included the original complete gel in the source data file as requested.

Reviewer #2 (Remarks to the Author):

*The manuscript by Wilson and colleagues describes structural and functional features of tripartite pores formed by a proteins AhIA, B and C from bacteria *Aeromonas hydrophila*. The authors have used bioinformatics approach in order to search for novel representative of cytolysin A (ClyA) family of pore forming toxins by using a known member as a probe. They have discovered a representative in *A. hydrophila* and have found additional components in its genome. They have expressed all three components in *E. coli* and performed structural and functional analysis with recombinant proteins. Based on the structural and functional data they propose a model of pore assembly, which is a novel for the ClyA family.*

The novelty of the presented work lies in the fact that the pore is formed by the three components and not one (as in the case of ClyA) or two (as in the case of YaxAB). The structural features of AhIA and B components that are studied here show great resemblance to other known members of the ClyA family, with some exceptions, so this is not particularly novel aspect of the work. The authors also show some TEM data, but they are not of the sufficient quality to draw any firm conclusion about the pore structure. In fact, the positioning of AhIA component in the pore and the model of AhI pore assembly is not adequately supported by the presented data and requires further work.

19. We thank the referee for their careful reading of the manuscript and have completed a number of extra experiments as suggested:

In the original manuscript we showed using lytic assays that maximal lysis of erythrocytes occurs only in the presence of all three AhI PFT components. We have repeated the EM images using erythrocytes as well as liposomes (Lines 212-217), liposome float assays (Lines 219-335), time course lytic

assays (Lines 194-217), and further negative stain EM of the end points of both of these assays and undertaken a more extensive modeling analysis of AhIA (Supplementary Figure 15; lines 519-554).

The liposome float assays show that AhIA, AhIB and AhIC all bind to liposomes when added together. Negative stain EM images of the floated liposomes show the presence of pore structures (Figure 1, lines 211-217), which have a larger diameter (16 nm) than the AhIB pores alone (10 nm). In addition, ultracentrifugation of a mixture of AhIA, AhIB and AhIC with detergent shows that assemblies > 400kDa containing all three components are formed.

We have also made a more thorough sequence and modeling comparison of AhIA to the structures of the other family components (Figure 7, Supplementary Figure 15; Lines 519-554). These comparisons of AhIA show that it most likely has a very similar structure to NheA and AhIB, with the same overall fold as the structures of the other ClyA family components in each system. Like these components, AhIA most likely will undergo a similar conformational change from soluble to pore structure. Sequence analysis shows that the proposed membrane traversing component of AhIA (α 3- α 4) would be constructed from two helices, one amphipathic and one hydrophilic. Modelling studies show that AhIB and AhIC could form a AhIBC pore in a similar structure to that seen for YaxAB and XaxAB (Figure 6, 7, Supplementary Figure 15; lines 485-517). However, this AhIBC pore would have a hydrophobic lining to the pore, unlike all other ClyA family member pores, that have a hydrophilic pore lining. We have modeled AhIA into a AhIBC pore to form a AhIABC pore that provides the hydrophilic lining to the pore, that is expected based on the similarity of the family structures overall. Taken in combination with the assays, which show that AhIA is required for maximal lysis, these new experiments are compelling evidence that AhIA must be a component of the fully lytic pore.

Comments:

20. *Line 147: Supplementary table 1 is just the list of proteins identified after the Blastp search by using AhIB as a probe. The results for AhIA and C are mentioned (lines 143-144), but actual sequences are not presented. I propose to put hits for all three components in Supp. table 1 and organise this table appropriately.*

We agree and have added BlastP hits when searching with AhIA and AhIC, and have included the sequence alignments in Supplementary Table 1.

21. *The SEC data on Supplementary Figure 2 shows additional peak at app. 42 ml for B and C and this peak, quite significant, is left unaccounted for by. Also, it is not clear which fractions are shown on the SDS-PAGE gels, is this peak included? What does the CFE and Ni-HP mean on the gels? The molecular weight markers are used but not specified to which molecular weights bands correspond to. The chromatograms and gels should be better annotated.*

We agree that Supplementary Figure 2 is somewhat confusing and have altered both the figure to annotate the SEC traces and gels more clearly, and the legend to reflect these changes. The SEC peak at 42ml arises from light scattering of aggregates in column void volume, not from UV absorption of soluble protein.

22. *It seems that the B component is different in a and c panels. While it is quite pure in peak2 fractions in a, it has a lot of impurities present in panel c gel.*

Please explain.

In Supplementary Figure 2c all samples have been boiled before loading on to the SDS-Page as described in the methods section (lines 6303-635). AhIB partially forms a large aggregate when boiled (top band). The lower band is the monomeric AhIB, no stack was used on this gel and so the lower band is smeared and not concentrated. The Figure has been relabeled to make this clear and the legend has been altered to reflect this (Supplementary Figure 2).

23. *This Figure also shows the result for 1:2 mixture of B and C components, but the ratio between the peaks does not corresponds to 1:2 ratio (presuming they absorb to a similar extent). This should also be clarified.*

The relative absorbance of AhIC to AhIB is very low because AhIC does not contain any Trp residues and thus the extinction coefficient is very small ($1490 \text{ M}^{-1}\text{cm}^{-1}$) compared to that of AhIB ($25440 \text{ M}^{-1}\text{cm}^{-1}$). We have changed the Figure legend to reflect this (Supplementary Figure 2).

24. *Panels c and d are mixed in Figure 1.*

This has been corrected and the figure changed extensively to add the results of the new experiments.

25. *Line 186: here also Supplementary Figure 3 should be cited. Now it is left unaccounted for by. In general, some supplementary Figures are not cited in order they appear.*

The Figures and supplementary Figures have been reordered to follow the text order.

26. *Line 190- what is the reason for AhIA degradation in presence of detergents?*

The purification and stabilization of AhIA required the presence of 50mM Arg and Glu in the buffer (lines 628-630). In the UC detergent assays, no amino acids were added to the detergent buffer, and therefore AhIA may well be degrading slightly, resulting in extra lower molecular weight bands on SDS-PAGE, this has been added to the text (lines 681-684).

27. *It is not clear why there is a need to use detergents in experiment presented in Figure 1e (line 195). The detergent micelles are poor substrate for membrane environment, presuming the reason was to test the ability of proteins to bind to lipid membranes. Why were not appropriate lipid vesicles used, which could better report the ability of individual components to associate with lipid membranes in isolation or when in complex. I think this panel has low relevance to the in vivo situation.*

We agree with this comment. The ultracentrifugation assays in the presence of detergent show that large molecular weight complexes are produced in a hydrophobic environment from mixtures of AhIA, AhIB and AhIC or AhIB and AhIC and from AhIB alone, but not from AhIA or AhIC in isolation.

To complement the ultracentrifugation assays, we have undertaken new liposome float assays, which show that all three components interact with the lipid bilayer together and individually. Negative stain EM images of the liposomes from the float assays show pores with AhIB, and much larger pores with AhIBC or AhIABC, the same sizes as the pores seen when the components are incubated with erythrocytes.

As we have shown that AhIA cannot form a large complex alone, it must be binding to the bilayer and interacting with AhIB and AhIC in the AhIABC pores (Figure 1, lines 219-235; 194-217).

28. *To better understand explanations of the structure of pre-pore AhIB in the text I propose to include another panel on Figure 2, where different parts of the pre-pore AhIB (i.e. helices, head, tail etc.) would be highlighted, similar to the one shown for the soluble AhIB on Figure 5a.*

This Figure has been altered as requested and is now Figure 3.

29. *Line 201: on what basis the authors conclude that the solved structure of AhIB is a membrane bound form. Without showing some evidence this should be rephrased.*

The AhIB pore dimensions in erythrocytes from the EM images match those of the crystal structure (lines 272-280). In addition, the new liposome float assay shows that AhIB interacts with the lipid bilayer and the EM of these liposomes show pores of the same dimension have been produced (Figure 1, lines 219-235). We have altered the text to read “the structure of an oligomeric pore form” (line 267).

30. *Lines 213 and 214: alpha helices 3 and 4 could also be labeled in the left image of panel a (Figure 2). Also the head and tail regions of the protein, which are referred to in lines 2019 and 211, could be highlighted in the panels of Figure 2a.*

These changes have been made (Figure 3).

31. *The features of soluble AhIB, in particular the beta-tongue with hydrophobic residues should be compared with other members of ClyA family. Is there a common mechanism present in the whole family? The comparison could include structural models from different representatives with the hydrophobic part labeled, possibly as a supplementary Figure.*

A new Figure has been produced (Figure 2), which compares the structures of soluble CylA, AhIB, HblB, MakA and NheA, and has been discussed in the text (lines 251-256).

32. *Line 321: it is claimed that no membrane-bound assemblies of AhIC alone could be observed in electron micrographs. Why are these data not presented in Figure 4?*

These images have been added to Supplementary Figure 3.

33. *The observation that some of liposomes are saturated with pores and others remain free of pores (lines 328-329) requires an explanation. Could the reason be high cooperativity of the membrane assembly process of AhI pores?*

Although the reason for this phenomenon is unclear, similar saturation of liposomes with other ClyA family members have been observed. One reason could be high cooperativity in the system, and we have discussed this in the text (lines 237-245, Figure 1).

34. *The quality of the presented data (Figure 4) does not clearly show that tetramer of AhIA cannot bind the membrane. I.e. the proposal that AhIA tetramer must dissociate in order to bind the membrane (line 378) should be supported by some functional data.*

We presume the referee is referring to AhIC in this comment. We have used both glutaraldehyde and EDC to cross link the soluble tetrameric form of AhIC to produce a sample of exclusively AhIC dimers, using this in assays abolishes lytic activity (Supplementary Figure 11, lines 381-388). When AhIC was cross linked with glutaraldehyde alone, SDS-PAGE showed an equal mixture of dimers and monomers. Using this partially cross-linked AhIC in

haemolytic assays resulted in a greater than 50% loss of activity. This shows that monomeric AhIC is required for lysis and thus the tetramer must dissociate for the pore to form. This has been discussed in the text (lines 387-390).

35. *Lines 392-399 explain the results of the functional mutants assay. This should better fit to the results section.*

This section has been moved to results (lines 388-398).

36. *Why were E.coli lipids used for proteoliposome preparation? Is this a natural substrate for an AhI system? Is there any lipid specificity associated with the binding to the lipid membrane by any of the AhI components?*

E.coli lipids were used as they form cleaner samples for TEM analysis. However, we have repeated all the EM analysis using erythrocytes, the results of which are consistent with those seen with liposomes, showing that the liposomes are a good bilayer model (Figure 1).

We have not identified any lipid specificity for the AhI PFT system.

37. *Where is evidence that pores in Figure 4e are formed by all three components a, b and c? Without further analysis of images and additional structural details of the pore it is not possible to draw any firm conclusions. Is there any functional data consistent with the model proposed in Figure 8?*

The authors could try using size markers such as fluorescent dextrans in order to get an idea about the pore diameter of various AhI assemblies, or perform ionic current measurements in a planar lipid membranes setup.

We have undertaken new experiments: liposome float assays (Lines 219-235), time course lytic assays (Lines 194-217), and further negative stain EM of the end points of both of these assays and a more extensive modeling analysis of AhIA (Figures 1, 7, Supplementary Figure 15; lines 516-547).

The liposome float assays show that AhIA, AhIB and AhIC all bind to liposomes when added together. Negative stain EM images of the floated liposomes show the presence of pore structures (Figure 1), which have a larger diameter (16 nm) than the AhIB pores alone (10 nm; lines 213-217). EM images of floated liposomes with AhIA or AhIC alone do not show any pores, and no pores at all are seen in all of the fractions without liposomes. In addition, ultracentrifugation of a mixture of AhIA, AhIB and AhIC with detergent shows that assemblies > 400kDa containing all three components are formed, but that AhIC or AhIA alone do not form large assemblies. The time course kinetic assays show that AhIC primes the membrane for pore assembly and that AhIA needs to be added at the same time as AhIB for fully active pores to be formed (Figure 1, lines 194-217).

We have also made a more thorough sequence and modeling comparison of AhIA to the structures of the other family components (Supplementary Figure 15, lines 519-554). These comparisons of AhIA show that it most likely has a very similar structure to NheA and AhIB, with the same overall fold as the structures of the other CylA family components in each system. Like these components, AhIA most likely will undergo a similar conformational change from soluble to pore structure. Sequence analysis shows that the proposed membrane traversing component of AhIA (α 3- α 4) would be constructed from two helices, one amphipathic and one hydrophilic. These two helices could pack together to form a membrane-spanning region of overall amphipathic character. Modelling an AhIABC pore using the structures of YaxAB and XaxAB as a guide shows that α 3- α 4 of AhIA could pack against the hydrophobic surface of AhIB to give an hydrophilic pore lining, as seen in all other family members. This model would require a larger diameter of the

membrane spanning head region, but not of the extracellular tail than the YaxAB pore, and would thus conform to the pore dimensions obtained from EM (Figure 1, lines 213-217). Taken in combination with the assays, which show that AhIA is required for maximal lysis, these new experiments are evidence that AhIA must be a component of the fully lytic pore.

Reviewer #3 (Remarks to the Author):

The manuscript describes a novel tripartite alpha pore-forming toxin (PFT) system, AhIABC, with functional and structural analogy to the already described ClyA type toxin. The authors present structures of two out of three components of the toxin system, with the membrane-permeating component in two conformations. Based on structural and functional data and also a site-directed mutagenesis study, a model of holotoxin pore assembly is proposed. The presented data support a pore formation reaction analogous to the described unimolecular or bimolecular alpha-PFTs. The authors demonstrate well that only a holotoxin made up of all three components form a fully functional pore, underlining the novelty of their finding. Altogether, the results and the novelty of the three-component alpha-PFT make the paper interesting for a publication in Nature communication.

The fact that five different structures (where 2x 2 show the identical state, if I'm not mistaken) combined with the fact that two conformations are present in the AhIB pre-pore make the manuscript confusing. I recommend to present the individual structures in a more accessible way by clearly writing what the two different structures of the AhIB pre-pore show, what they have in common and what are the differences (if any), analogous to the two AhIC structures. The storyline often goes forward and back in the Figures, which makes it a bit difficult to read. That could be improved.

Furthermore, the pore assembly model should be strengthened by more functional and mutational studies, please see below.

Altogether, when the manuscript would be complemented with the suggested experiments to strengthen the assembly hypothesis and the storyline would be presented in a clearer way, I recommend publication.

38. Suggested additional experiments to strengthen the content of the paper

- (1) Hemolysis kinetics assays (see. l.167ff and Fig.1) to get more information about the cell lysis behavior of individual components, especially after pre-incubation of non-lytic components with erythrocytes and addition of other components of the holotoxin. Is the activation then fast or slow, with or without a pronounced lag time? That would give information about the timing of assembly and conformational transitions. The experiments could also help to support or disprove the assembly model with AhIC as initial membrane-bound subunit.*
- (2) The UC experiments show only that large assemblies that precipitate in an UC run are formed. Evaluation of that complexes via gel filtration and electron micrographs would give more information. Are similar complexes formed like observed in Fig. 4? (see l.182ff).*
- (3) Site-directed mutagenesis: Mutagenesis experiments on the AhIC head domain showed the loss of hemolytic activity. Mutagenesis on the hinge regions of AhIB would strengthen the assembly and conformational transition model, analogous to experiments described*

for YaxB and ClyA.

We agree with the referee that extra experiments would strengthen the content of the paper. As suggested, we have thus completed a series of haemolytic assays, both as time courses, and following pre-incubation with the different components of the AhI PFT system. We have also completed a series of lysosome floatation assays to complement the ultracentrifugation experiments, analyzing the different assays by EM to show whether and what type of pore structures are formed. In addition, we have repeated the EM analysis of the pore structures using erythrocytes rather than liposomes as the lipid bilayer component, which show that the structures described using lysosomes are the same as those seen in erythrocytes. Haemolytic assays using cross-linked AhIC support the model of pore formation, as lysis is abolished if the soluble AhIC tetramer is prevented from disassembling into monomers, to expose its hydrophobic head domain. The results of these extra experiments are described below.

Point-by-point analysis of the manuscript:

Introduction:

39. l. 57: *“initially fold into”*: better write *“are produced as”* to avoid confusions
Text altered as requested (line 56).
40. l. 76: *A recent publication also compares the XaxAB pore with the respective monomer structures. The XaxAB pore shows more details than the YaxAB pore and stresses the fact that pores with different assembly numbers can be created.*
The XaxAB structures had not been published when we submitted our manuscript. We agree that they are important in describing the ClyA family and should be included in the revised manuscript. We have changed the introduction (lines 75-89) and also referred to these structures throughout the discussion and figures as appropriate (see reply to comment 11).
41. l. 109: *“the AhIC tetramer dissociates”*: *It is unclear to me what is meant here, where or from what it dissociates. Please write to be more clear.*
We are describing the dissociation (disassembly) of the tetramer of AhIC, so that the hydrophobic elements become exposed and to interact with the lipid bilayer. The wording has been changed to emphasize this (lines 116-124).
42. l. 110: *“large scale conformational change”*: *How large? A brief comparison with the changes of ClyA and YaxB would be helpful here.*
The conformational changes in ClyA and YaxB have been included in the description (lines 116-124).
43. l.111: *modeling of AhIA: That suggests a homology model that is presented together with the crystal structure. However, this is not the case here.*
This sentence has been reworded to be more clear (lines 121-124), we have also undertaken a more extensive modeling of AhIA, which is described in the text (Supplementary Figure 15, lines 519-554) .

Results:

44. *General arrangement of description of structures and Figures:*

I find it a bit confusing to start with the AhIB pre-pore, then show them on the membrane and then describe the monomer structure. References to Figures are then going from 2 to 3 to 4 and then back to 3. Maybe re-arrange the Figures 3-5 to follow the storyline better.

The paper has been re-ordered, so that the soluble structure of AhIB is described first, then the pore structure of AhIB. The Figures have been rearranged to follow the text order.

45. *I.144ff and SI Table 1: In which homology group was the single component ClyA found? Please state in the text, if not found at all, please indicate why it was not identified as a homolog.*

The sequence homology between ClyA and AhIA, AhIB or AhIC is very low (15.1, 15.2, 16.4 % identities, respectively) and thus alignments to ClyA were below the cutoff used for the table and were not identified by any of the BLAST searches (E values of 3.7, 0.8 and 2.2 against AhIA, AhIB, AhIC, respectively). The observed similarity across the ClyA family is seen in the structures of the soluble and pore forms the pattern of hydrophobic residues and the architecture of the pores.

46. *I.166f: better wording in my opinion would be: "..., indicating that indeed all three components of the AhIA/AhIB/AhIC toxin system are required for its maximum activity."*

The sentence has been re-worded as requested (lines 177-178).

47. *I.167 and Fig1: An experiment that does not only show the end product after a distinct incubation time, but an on-line measurement of lysis kinetics (like described for ClyA) would give us more information regarding the lysis behavior with the different components. It would also enable the possibility to pre-incubate erythrocytes with different components and add the 3rd component later. If e.g. there is no lag time anymore when adding AhIA to pre-incubating AhIBC, it would show that no large conformational change is necessary for that component.*

The haemolytic assays with erythrocytes have been repeated using lower concentrations of toxin components, to slow the lysis, enabling time course experiments to be undertaken (Figure 1 and lines 194-217). These experiments show that if erythrocytes are pre-incubated with AhIC, maximal lysis occurs immediately after addition of AhIB and AhIA, whereas there is a time lag when all three are added together to the erythrocytes. If the erythrocytes are pre-incubated with AhIB, prior to addition of AhIA and AhIC, or AhIC alone, lysis is retarded, and even after 40 minutes has not reached the level obtained when these components are mixed together without any pre-incubation of erythrocytes. This suggests that AhIA can only add very slowly to AhIB or AhIBC pores, or perhaps that AhIABC pores are formed from remaining soluble AhIB and AhIC that had not associated with bilayer. In addition, full saturation of liposomes or erythrocytes is only seen when all components are incubated together, or the bilayer is pre-incubated with AhIC prior to AhIB and AhIC addition. Similarly, maximal lysis does occur for cells pre-incubated with AhIA prior to the addition of AhIB and AhIC, but AhIB/AhIC alone only exhibit 50% of the lysis of all three components.

48. *I.174ff: Figure1d for AhIC, Figure1c for AhIB?*

Figure 1 has been substantially altered and corrected, as requested.

49. *I.171ff: What happens if AhIA, B or C are used in excess over the other*

components? Can you show also data points available for them? Do the curves for AhIA and AhIB reach saturation at a 1:1:1 ratio or is there a further increase (or even decrease, like NheABC)?

These experiments have been completed and show that the curves reach saturation at 1:1:1 ratio, with no increase or decrease in lysis when components are introduced in excess. The text has been altered to describe these experiments (Figure 1, lines 189-192).

50. I.180: Please describe briefly what happens in NheABC and speculate why AhIABC is different.

NheC and NheB form a complex in solution, which sequesters the two components, preventing them assembling into a lipid puncturing pore, thus decreasing lysis. YaxAB can also assemble into a complex in solution, but this is not believed to inhibit lysis. In contrast, no complexes of AhI components can be observed in the absence of bilayer or detergent, indicating that pore formation only occurs on the membrane for the tripartite AhI toxin. We have altered the text to reflect this (lines 169-175).

51. I.182ff, UC experiment: In that experiment it is only shown that the proteins are in the pellet at high-speed centrifugation, nothing more. As you have the possibility to visualize the samples in negative staining, do you see large symmetric assemblies that are consistent with a possible pre-pore or the pore? Gel filtration on Superose 6 could also give more information regarding the size of the complex, if it is aggregate or a species with a defined size. IN Fig. 1e it is also obvious that only ~20% of the trimeric complex are in the pellet, while about 50% of the BC complex are in the pellet. Any possible reasons why?

I recommend additional experiments (neg stain EM, gel filtration) to specify the type of large complexes.

To aid interpretation of the UC results we have undertaken liposome floatation assays and analyzed the structures present by negative stain EM for both the UC and float assays experiments. This makes a more complete analysis of the pore formation, which is presented in figures (Figure 1, Supplementary Figure 3) and text (lines 219-235).

These extra experiments show that AhIA, AhIB and AhIC all bind to the liposome lipid bilayer. Pores are only observed for AhIB, AhIBC or AhIABC. The pores formed from the AhIBC and AhIABC complexes are much bigger (16nm) than those observed for AhIB alone (10nm).

In the liposome float assays, similar amounts of AhIB and AhIC bind to the liposomes independent of the amount of AhIA present. As each component can bind to the bilayer alone, this suggest that in the AhIABC float assays both pores and individual components could be bound to the liposomes. In the ultracentrifugation experiment, however, only species > 400kDa pellet and thus the experiment with AhIA, AhIB and AhIC, where all of the AhIA is pelleted with a similar amount of AhIB and AhIC, perhaps indicates that only AhIABC pores have been formed. The difference between this trimeric complex UC assay and the AhIBC UC assay suggests that only AhIABC pores are formed if all three components are present.

52. I.210ff, indicate neck, tail etc. in Fig.2; state what is outer and inner diameter of the pore

Changes made to Figure as requested (Figure 3).

53. I218: indicate Phe and Tyr in the same Figurepanel (Y245 is indicated in

Figure2A, F203 in Figure2B)

Changes made to Figure as requested (Figure 3).

54. I.221ff: *Two conformations of AhIB in the oligomeric structure and two different AhIB (pre-)pore structures solved – that is confusing. What is the difference between both crystal structures? Why is one structure named pre-pore and the other one pore? Please make that clear.*

We agree this is confusing. Both AhIB oligomeric structures are now referred to as pore structures, rather than pre-pore and pore. The packing of the head domains are different between the two crystal structures and these differences are discussed in the text (Supplementary Figure 6, lines 298-301).

55. I.247ff: *The AhIB structure's head looks strikingly similar to the ClyA monomer and also the HblB structure. Please refer to that and maybe compare them briefly. It might be useful for a better understanding of the toxin system to stress here that AhIB is that part of the trimeric toxin that is most similar to ClyA.*

A new figure comparing the soluble structures of ClyA, HblB, AhIB, MakA and NheA has been presented (Figure 2), together with additions to the text (lines 251-256).

56. I.280ff.: *Can you provide information that the arrangement of the tetramer in the crystal package is consistent with the real situation (e.g. cross-linking studies or negative stain images)? How would the tetramer dissociate in pore formation? Are there examples in the literature where a Leu-Zipper is dissociated by membrane contact interactions? Does the tetramer dissociate in a detergent solution?*

AhIC seems to be the toxin subunit that is consistent with The A subunit of binary toxins, please indicate that here. Also an additional Figure panel comparing YaxA and AhIC (as well as YaxB and AhIB) would be helpful. SEC analysis of soluble AhIC shows that only the tetramer is present in solution (Supplementary Figure 2, lines 341-343). We have crystallised AhIC in two different space groups, in both of which the crystal is composed of an array of tetrameric AhIC. In one crystal form the asymmetric unit contains a dimer, with crystal symmetry forming the tetramer, whereas in the other space group the asymmetric unit contains a complete tetramer. In addition, we have now determined the structure of the triple mutant of AhIC, which also forms tetramers in the crystal. However, these tetrameric assemblies are too small to be clearly visible in the negative stain EM. We have cross-linked soluble AhIC using glutaraldehyde and EDC, to produce dimers of AhIC, which when used in assays abolish lysis (Supplementary Figure 11 and lines 381-398). When glutaraldehyde is used alone SDS-Page analysis reveals an equal mixture of dimers and monomers, and in assays using the cross-linked AhIC, lysis is reduced by >50%, as expected based on the amount of monomeric AhIC available. These results combined with the structure of AhIC, which shows the hydrophobic head buried in the tetramer, show that disassembly of the tetramer has to occur before pore formation. SEC analysis in the presence of detergent was inconclusive, as the detergent interferes with the column.

We have compared AhIC with YaxA and XaxA, (Figure 5, Supplementary Figure 7; lines 374-379), highlighting the structural and functional similarity between AhIC and YaxA and XaxA.

57. I.290: *How significant is the subunit difference in AhIC in comparison to AhIB?*

A discussion on the similarity between AhIC and the two pore conformations

of AhIB has been added to the text (lines 345-356), together with a new Figure highlighting the similarities (Figure 6).

58. *I.299: K151 is not shown in the Figure.*

Figure changed as requested (Figure 5).

59. *I.329: To underline the finding that there are liposomes with and without pores, an additional Supplementary Figure with several images of liposomes would be helpful.*

Additional supplementary figure added (Supplementary Figure 3e).

Discussion:

60. *I.356ff: A supplemental Figure comparing the conformational transitions of ClyA, YaxB and AhIB would be helpful.*

A figure showing these transitions has been added (Figure 4).

61. *I.392ff: The chapter describes experimental results, I think it would be better in the "Results" section.*

Agreed, and this section has been moved to results (lines 388-398).

62. *I.434ff, model of toxin assembly: If the model is correct, it should be possible to identify AhIC bound to the membrane alone, either via EM by analyzing liposome and soluble fractions in solution via SDS-PAGE (analogous to Fig.1e). Are EM images or biochemical experiments showing the co-localization of AhIC with membranes available to show?*

The new liposome float assays show that AhIC can bind to a lipid bilayer and the time course lysis assays show that pre-incubation with AhIC removes the time lag when no pre-incubation occurs (Figure 1, lines 194-235).

63. *I.463ff: It should also be stated here that the AhIABC pores appear larger than the AhIBC pores. The side view in Fig.4e is definitely larger than the top views in Fig. 4d suggests. If possible, a low-resolution 3D reconstruction of more images could be compared with the YaxAB or XaxAB pore to see size differences and also show the presence of an additional molecule.*

The size of the AhIABC pores visualized in negative stain EM (16 nm diameter, average of 100 pores) are larger than the AhIBC pores (10 nm, n=100; lines 212-216). In addition, these pores are of a comparable diameter to the cytosolic tail end of the widened funnel pore of YaxAB and XaxAB. As we now propose in the discussion (lines 541-554, Figure 7), an AhIABC pore would have a tail diameter only slightly larger than that of an AhIBC, but with a widened membrane spanning head. As the pore is funnel shaped, only the large tail dimension can be seen in negative stain EM.

64. *L468ff. A Figure showing NheA and the proposed domains interacting with the AhIB analogon and the TM regions would be helpful. Does the sequence identity allow the building of a homology model of AhIA?*

The soluble NheA structure has been compared to soluble AhIB (Figure 2, lines 251-256). It is difficult to model the interacting regions between AhIB and NheA in the pore, as no pore form structure of NheA is known. However, we can align the sequence of AhIA onto the soluble structure of NheA.

Assuming that AhIA undergoes a similar conformational change to that seen in AhIB and ClyA on pore formation, the membrane spanning region of AhIA can be identified. This is shown in both the helical wheels and as a Phyre 2

homology model (Supplementary Figure 15, lines 519-536). The hydrophobic residues on the amphipathic helix ($\alpha 3$, TM1), which pack against AhIB TM helices in our model of AhIABC are either identical or very highly conserved in all the Gram negative PFTs we have identified (Supplementary Figure 15) indicating that this surface is most likely involved in packing of the membrane spanning component of the pore.

65. *1.489ff: The last chapter of the discussion gives a very good overview of the differences and common features of the mono- bi- and tripartite toxin complexes. It could be complemented by a Figure showing schemes of the individual components as monomers and assembled.*
This Figure has been added (Figure 7).

Methods:

66. *1.563ff: Something seems to be missing in that sentence, does it mean AhIA was stored in buffer with 50mM Arg and Glu?*
Yes, text altered (lines 628-630).
67. *1.572ff: Were the non-Se-Met expressions also performed in minimal medium? It is not stated in the section above.*
Sulphur-met expressions were performed in LB, text altered to make this clear (lines 616).
68. *1584: Co-purification as heading is misleading, better: "Complex formation analysis"*
Agreed and changed (line 649).
69. *1.602: Is there any reason why a 1% erythrocyte suspension was used for the AhIC head mutant instead of 0.25%? Are the results comparable with respect to stoichiometries of erythrocytes and toxin?*
All assays have been repeated using the same concentrations of erythrocytes, which show the same results (Supplementary Figure 11, lines 667-670).
70. *Crystallization: To be consistent, maybe also describe crystallization and structure determination of the AhIB monomer before the oligomer, like it was done for AhIC.*
This has been done in the results section to aid flow, however, in the methods section we have described the Se-met AhIB pore form structure solution first, as it was the model from this structure that was used in the molecular replacement solution for the soluble form.
71. *For AhIB monomer, crystal growth take a month – how long did it take for the others?*
Crystals for all the different AhIB structures grew at similar rates, whereas AhIC crystals grow in a week. Details added to methods section (lines 722, 740, 775).
72. *What re the differences between the two AhIB pore structures?*
The structural differences are now discussed in the results section (Supplementary Figure 6, lines 271-273; 298-302), which mostly relate to the different crystal packing of the end of the head domains.

Figures:

73. *Fig1 b-d: Panels c and d are referred wrongly in the text.*
Agreed, this Figure has been changed substantially.
74. *Fig. 2: The same colors for the same type of protomers and the lack of depth of focus make it difficult to identify and trace different protomers, especially in the left and middle view. Please improve the Figure, e.g by using color gradients for the protomers (e.g. light green to dark green and light orange to brown for the two types of protomers). Also shadows, border lines to the models and depth of focus like in the right panel would be helpful. In panel b, please indicate which views are shown (section/side view). A panel comparing the two types of protomers side-by-side is missing and would be helpful. It would be also helpful to indicate the different regions and crucial residues in that panel (text l. 210ff).*
The figure has been altered to reflect these suggestions (Figure 3).
75. *Fig.3: color the two hinges with different colors, show the Figure after the individual Figure of the AhIB monomer (Fig.5).*
Change made (Figures 2, 4).
76. *Fig.4: It would make it easier for the reader to indicate the contents of the liposomes also in the Figure itself, not only in the legend. Indicate pore top views also in panels d and e. It might also be better for understanding the storyline of the text to split that Figure in parts that complement the respective Figures of the individual components.*
This Figure has been changed. We have compared the liposome EM and erythrocyte EM images in Figures 1, 3. Magnified parts of each image have been split away with the relevant enlarged EM images for each component shown in the appropriate place. (Figure 1,3).
77. *Fig.6: re-arrange, like the storyline is described in the text (first description of AhIC, then comparison with AhIB). A scheme of the Q subunits is missing in panel B.*
This has been changed in Figure 5.
78. *Fig.7: b,c: Is one AhIB replaced by an AhIC? Please indicate that in the text and Figure legend, ans also label the subunits in the Fig. panel.*
This has been changed (Figure 6).
79. *Fig.8: AhIA missing in the scheme in b*
What's the physiological relevance of the AhIB "pre-pore"? For me it appears as an artificial off-pathway construct that cannot be recovered to form a fully active AhIABC pore. Therefore, the term "pre-pore" is misleading.
A side view of the proposed AhIABC pore in the membrane would be helpful to understand the positioning of AhIA.
A new schematic has been prepared that includes AhIA (Figure 7). We agree with the statements made above and our new kinetic lysis experiments confirm that the inactive AhIB pore can only be recovered into an active pore over a long period of time. We assume this is due to partial (or complete) disassembly of AhIB pore, followed by reassembly with the other components.
80. *Supplementary 2: Indicate the sizes of the proteins in panel C. It looks like the mass difference between AhIB and AhIC is much more than only 8 kDa*

because of the many marker proteins between them. Contrast of gels in a and b is very low. Indicate marker protein sizes in all gels.
All agreed, Figure relabeled (Supplementary Figure 2).

81. *Supplementary 5: AhI instead of Alh. Use colors with more contrast for map and model, also in Suppl. Figure4a,b. IN both Suppl. Figure4 and 5, please indicate from which part of the structure the maps are shown.*

Changes made as requested (Figure now Supplementary Figure 4, 9).

82. *Suppl. 6: An additional side view showing where the disordered density is located would be helpful.*

A new Figure has been prepared and included (Supplementary Figure 8).

83. *Suppl. Fig. 10ff: 2x Figure 11? The overlays are sometimes difficult to interpret. In my opinion it would be more helpful to have the two structure schemes not overlaid, but side-by-side.*

A new Figure has been created in this format (Figures 7, 13).

Tables:

84. *Table 1 resolution: Some values are given as low-high, others high-low. Please change them to be consistent.*

This table has been changed to be consistent.

REVIEWERS' COMMENTS:

Reviewer #1 (Remarks to the Author):

The authors addressed all my concerns.
Best regards.

Reviewer #2 (Remarks to the Author):

This version of the manuscript is much improved, based on the comments from reviewers. The authors have added more data, reorganized the manuscript and add new figures in order to generate much better description of this interesting pore forming system. They have satisfactorily addressed all concerns and I do not have any further comments.

Reviewer #3 (Remarks to the Author):

The additional experiments and the additional site-directed mutagenesis data complement the manuscript well. All my comments and suggestions were addressed properly. The figures are now presented in a more clear way, making the manuscript more accessible. Therefore, I'm happy to recommend the manuscript for publication.

I would however ask the authors to address some minor issues:

Figure 1b: The fact that already a sub-stoichiometric concentration of AhIC is enough to cause 100% activity should be discussed in the main text. This could indicate that AhIC functions as receptor, and that not all AhIB/AhIA need an AhIC to bind to cell membranes.

Figure 1e (and SI Fig 3b): Please indicate from where in the figures the insets were taken.

Figure 1f: What do the individual fractions show? Is it the gradient from top to bottom, from the left to the right in the gel, or are it different conditions? Please indicate. Same for SI Fig. 3a

Supplementary Figure 3 d,e: Please indicate the protein aggregates or pores, if applicable.

Reference: **NCOMMS-18-23936**

Response to reviewers:

REVIEWERS' COMMENTS:

Reviewer #1 (Remarks to the Author):

The authors addressed all my concerns.
Best regards.

Reviewer #2 (Remarks to the Author):

This version of the manuscript is much improved, based on the comments from reviewers. The authors have added more data, reorganized the manuscript and add new figures in order to generate much better description of this interesting pore forming system. They have satisfactorily addressed all concerns and I do not have any further comments.

Reviewer #3 (Remarks to the Author):

The additional experiments and the additional site-directed mutagenesis data complement the manuscript well. All my comments and suggestions were addressed properly. The figures are now presented in a more clear way, making the manuscript more accessible. Therefore, I'm happy to recommend the manuscript for publication.

I would however ask the authors to address some minor issues:

Figure 1b: The fact that already a sub-stoichiometric concentration of AhIC is enough to cause 100% activity should be discussed in the main text. This could indicate that AhIC functions as receptor, and that not all AhIB/AhIA need an AhIC to bind to cell membranes.

We agree with the referee that Figure 1b suggests that lytic pores might be formed with sub stoichiometric amount of AhIC, and we have added a sentence to reflect this (lines 492-495). We have already suggested that AhIC binding to membranes is the initial event in pore formation (e.g lines 432-443). The liposome float assays show that although all three components can each bind to membranes alone (Fig 1f), AhIC is required for any lysis to occur: AhIA and AhIB are not lytic in isolation and neither are AhIA/AhIB mixtures (Fig1a).

Figure 1e (and SI Fig 3b): Please indicate from where in the figures the insets were taken.

This has been done, with the zoomed in area highlighted.

Figure 1f: What do the individual fractions show? Is it the gradient from top to bottom, from the left to the right in the gel, or are it different conditions? Please indicate.

Same for SI Fig. 3a

A schematic of the experiment has been added to Figure 1f and SI Figure 3a and the legends changed to explain the fractions.

Supplementary Figure 3 d,e: Please indicate the protein aggregates or pores, if applicable.

Areas have been highlighted as requested.